# Characterizing the Predictive Impact of Modalities with Supervised Latent-Variable Modeling

**Divyam Madaan** [1]   **Sumit Chopra** [1 2]   **Kyunghyun Cho** [1 3]

## Abstract

Despite the recent success of Multimodal Large Language Models (MLLMs), existing approaches predominantly assume the availability of multiple modalities during training and inference. In practice, multimodal data is often incomplete because modalities may be missing, collected asynchronously, or available only for a subset of examples. In this work, we propose **PRIMO**, a supervised latent-variable imputation model that quantifies the **pr**edictive **i**mpact of any missing **mo**dality within the multimodal learning setting. PRIMO enables the use of all available training examples, whether modalities are complete or partial. Specifically, it models the missing modality through a latent variable that captures its relationship with the observed modality in the context of prediction. During inference, we draw many samples from the learned distribution over the missing modality to both obtain the marginal predictive distribution (for the purpose of prediction) and analyze the impact of the missing modalities on the prediction for each instance. We evaluate PRIMO on a synthetic XOR dataset, Audio-Vision MNIST, and MIMIC-III for mortality and ICD-9 prediction. Across all datasets, PRIMO obtains performance comparable to unimodal baselines when a modality is fully missing and to multimodal baselines when all modalities are available. PRIMO quantifies the predictive impact of a modality at the instance level using a variance-based metric computed from predictions across latent completions. We visually demonstrate how varying completions of the missing modality result in a set of plausible labels.

[1]Courant Institute School of Mathematics, Computing, and Data Science, New York University [2]Grossman School of Medicine, New York University [3]CIFAR LMB. Correspondence to: Divyam Madaan <divyam.madaan@nyu.edu>.

*Proceedings of the 43rd International Conference on Machine Learning*, Seoul, South Korea. PMLR 306, 2026. Copyright 2026 by the author(s).

## 1. Introduction

A central challenge in practical multimodal learning is the limited availability of all modalities for a downstream task. Many curated benchmarks, both in healthcare (Soenksen et al., 2022; Huang et al., 2025; Gu et al., 2025) and in standard multimodal learning (Antol et al., 2015; Goyal et al., 2017; Dancette et al., 2021; Tong et al., 2024; Liu et al., 2024; Yue et al., 2024; Wu & Xie, 2024), assume that all modalities are observed at training and inference.

In practice, modalities are missing for many instances, especially in healthcare, where paired data is often incomplete (von Kleist et al., 2023a;b; Erion et al., 2022). When a patient arrives at the hospital, only a limited set of measurements may be collected initially, and additional tests are ordered only when clinicians suspect a specific condition. This matters because acquiring additional modalities can be expensive and can pose risks to patients. For instance, in prostate cancer screening, MRI before biopsy can improve downstream decision-making, but it also adds cost and exposes patients to additional procedures and potential risks (Callender et al., 2021).

In these settings, the goal is not to fill in the missing inputs, but to understand what the missing modality would actually change for the prediction. This motivates the central question of our work:

> *For a given multimodal example,*
> *how does a modality affect the prediction?*

Most existing approaches model missing modalities as an imputation problem. They infer the missing modality conditioned on the observed modality and then treat the imputed value as observed. Many methods use generative models (Suzuki et al., 2017; Wu & Goodman, 2018; Shi et al., 2019; Sutter et al., 2020; 2021; Joy et al., 2022; Palumbo et al., 2023) to tackle this problem during inference by optimizing a variational lower bound on the data likelihood. This objective prioritizes reconstructing the input modalities; however, improved generative modeling does not necessarily translate into better discriminative performance. This is because there can be many ways to fill in a modality, and only some of them matter for prediction.

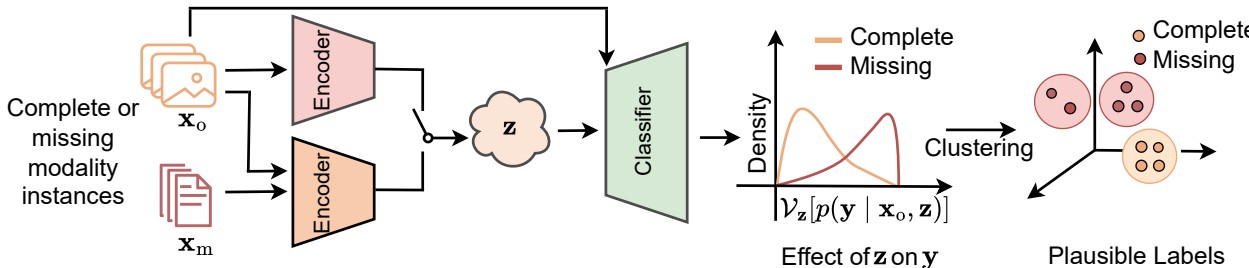

*Figure 1.* **Overview of PRIMO.** Given an observed modality $\mathbf{x}_o$ and an additional modality $\mathbf{x}_m$ that may be missing, PRIMO samples a latent variable $\mathbf{z}$ conditioned on the available modalities. The classifier maps $(\mathbf{x}_o, \mathbf{z})$ to predictions, and the conditional variance $\mathcal{V}_{\mathbf{z}}\left[p(\mathbf{y} \mid \mathbf{x}_o, \mathbf{z})\right]$ quantifies how changes in $\mathbf{z}$ affect the prediction. When both modalities are observed (orange), $\mathcal{V}$ is lower. When a modality is missing (red), $\mathcal{V}$ is higher. PRIMO then clusters the output logits across latent samples to visualize plausible labels under each availability scenario.

Mancisidor et al. (2024) partially mitigates this issue by incorporating a discriminative objective, but assumes fully observed multimodal training data. Other approaches discard partially observed examples and either use only complete training data (Suzuki et al., 2017; Wu & Goodman, 2018; Shi et al., 2019; Sutter et al., 2020; 2021) or make predictions only on fully paired data (Lee et al., 2023; Wu et al., 2024). None of these methods jointly optimize a discriminative objective while supporting partially observed modalities during both training and inference.

We thus need an approach that (i) uses both complete and partially observed examples during training and inference, and (ii) captures uncertainty in the missing modality pertaining to the predictions for each instance. The goal is not to produce a single value of the missing modality, but to characterize how different plausible versions of the missing modality would change the predictive distribution for a given instance. To achieve this, we propose **PRIMO**, a supervised latent-variable model that quantifies the **pr**edictive **i**mpact of any **mo**dality. At a high level, PRIMO measures the impact of each missing modality by modeling it as a latent variable. During inference, PRIMO draws many samples from the learned distribution over the missing modality to obtain a set of final predictions that captures the marginal predictive distribution and the uncertainty due to the missing modality (see Figure 1).

More formally, let $\mathbf{x}_o$ denote the observed modality, $\mathbf{x}_m$ denote the additional modality that may be missing for some instances, and a target label $\mathbf{y}$. Since modalities can be high-dimensional, directly modeling what part of $\mathbf{x}_m$ is relevant for $\mathbf{y}$ can be challenging. We thus use a continuous latent variable $\mathbf{z}$ to capture the information associated with the missing modality that is relevant for predicting $\mathbf{y}$. PRIMO is trained end-to-end to maximize the predictive distribution $p(\mathbf{y} \mid \mathbf{x}_o)$ when $\mathbf{x}_m$ is unavailable and $p(\mathbf{y} \mid \mathbf{x}_o, \mathbf{x}_m)$ when both modalities are observed. When $\mathbf{x}_m$ is absent during inference, $\mathbf{z}$ is sampled from a conditional prior $p(\mathbf{z} \mid \mathbf{x}_o)$. When both modalities are available, it is sampled from $p(\mathbf{z} \mid \mathbf{x}_o, \mathbf{x}_m)$.

This latent-variable formulation enables the characterization of predictive impact due to the missing modality for each instance. Particularly, we measure $\mathcal{V}_{\mathbf{z}}[p(\mathbf{y} \mid \mathbf{x}_o, \mathbf{z})]$ to quantify the effect of changes in $\mathbf{z}$ on the output predictions. Small values of $\mathcal{V}$ imply that the output is less dependent on the missing modality, whereas large values indicate a greater dependence. The distribution over logits yields instance-level estimates of modality impact and captures the range of plausible predictions induced by different latent samples. This also allows us to use PRIMO as a diagnostic tool in complete-modality scenarios to test modality dependence and identify when multimodal models rely on shortcuts (Fu et al., 2024; Tong et al., 2024; Madaan et al., 2026; Gu et al., 2025).

We evaluate PRIMO on synthetic and real-world multimodal benchmarks. These include a synthetic XOR dataset, Audio-Vision MNIST (Liang et al., 2021) with audio and vision modalities, and MIMIC-III (Johnson et al., 2016; Liang et al., 2021) with patient demographics and clinical time-series for mortality and ICD-9 code prediction. Across all datasets, PRIMO obtains performance comparable to unimodal baseline $p(\mathbf{y} \mid \mathbf{x}_o)$ when a modality is missing, and to a multimodal baseline $p(\mathbf{y} \mid \mathbf{x}_o, \mathbf{x}_m)$ when all modalities are available. Beyond predictive performance, PRIMO provides insight into the impact of different modalities for a given task. For example, we show that patient demographic information is sufficient for mortality prediction and neoplasm ICD-9 code prediction in MIMIC-III, while clinical time-series is essential for respiratory ICD-9 code prediction. The code is available at `https://github.com/divyam3897/PRIMO`.

## 2. Learning with Both Complete and Missing Modalities

We consider supervised multimodal learning where a modality can be missing during training and inference. For clarity, we focus on two modalities. Each example consists of an observed modality $\mathbf{x}_o$, an additional

modality $\mathbf{x}_{\mathrm{m}}$ that may be absent, and a label $\mathbf{y} \in \Delta^{C-1}$ over $C$ classes. The dataset contains complete examples $\mathcal{D}_{\mathrm{complete}} = \{(\mathbf{x}_{\mathrm{o},i}, \mathbf{x}_{\mathrm{m},i}, \mathbf{y}_i)\}_{i=1}^{N_c}$ and missing-modality examples $\mathcal{D}_{\mathrm{missing}} = \{(\mathbf{x}_{\mathrm{o},j}, \mathbf{y}_j)\}_{j=1}^{N_m}$. PRIMO learns a predictor that maps the available modalities to $\mathbf{y}$, using $(\mathbf{x}_{\mathrm{o}}, \mathbf{x}_{\mathrm{m}})$ when $\mathbf{x}_{\mathrm{m}}$ is present, and only $\mathbf{x}_{\mathrm{o}}$ otherwise.

To characterize the impact of a missing modality, our goal is not to reconstruct $\mathbf{x}_{\mathrm{m}}$ but to capture the uncertainty in $\mathbf{x}_{\mathrm{m}}$ that is relevant for prediction. PRIMO is a supervised latent-variable model trained end-to-end (Section 2.1) that supports both complete and missing-modality inputs. It samples latent completions from $p(\mathbf{z} \mid \mathbf{x}_{\mathrm{o}})$ when $\mathbf{x}_{\mathrm{m}}$ is missing and from $p(\mathbf{z} \mid \mathbf{x}_{\mathrm{o}}, \mathbf{x}_{\mathrm{m}})$ when both modalities are observed, which typically reduces predictive variance. This enables us to quantify how predictions vary across completions and use PRIMO for both prediction and modality impact analysis at inference time (Section 2.2).

## 2.1. Learning objective

We optimize variational lower bounds on the conditional log-likelihoods $\log p(\mathbf{y} \mid \mathbf{x}_{\mathrm{o}}, \mathbf{x}_{\mathrm{m}})$ and $\log p(\mathbf{y} \mid \mathbf{x}_{\mathrm{o}})$. Following the data generating process (DGP) in Figure 2, we model the label-relevant information in the missing modality $\mathbf{x}_{\mathrm{m}}$ with a continuous latent variable $\mathbf{z} \in \mathbb{R}^d$. We assume that $\mathbf{y}$ is conditionally independent of $\mathbf{x}_{\mathrm{m}}$ given $(\mathbf{x}_{\mathrm{o}}, \mathbf{z})$. Under this assumption, the predictive distributions for complete and missing-modality inputs are

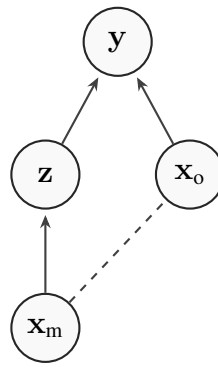

*Figure 2.* DGP for missing modalities. The dashed line denotes an a priori correlation between the two modalities.

$$p(\mathbf{y} \mid \mathbf{x}_{\mathrm{o}}, \mathbf{x}_{\mathrm{m}}) = \int p_\theta(\mathbf{y} \mid \mathbf{x}_{\mathrm{o}}, \mathbf{z}) \, p_\omega(\mathbf{z} \mid \mathbf{x}_{\mathrm{o}}, \mathbf{x}_{\mathrm{m}}) \, d\mathbf{z},$$
$$p(\mathbf{y} \mid \mathbf{x}_{\mathrm{o}}) = \int p_\theta(\mathbf{y} \mid \mathbf{x}_{\mathrm{o}}, \mathbf{z}) \, p_\omega(\mathbf{z} \mid \mathbf{x}_{\mathrm{o}}) \, d\mathbf{z}, \tag{1}$$

where $p_\theta$ is the predictive model and $p_\omega$ parameterizes the conditional latent distributions. Since these integrals are intractable, we introduce an approximate posterior $q_\phi$ and maximize the resulting evidence lower bounds (ELBOs) for both scenarios (see Section A of the Appendix for complete derivations).

**Case 1: Complete modalities** ($\mathcal{D}_{\mathrm{complete}}$). When both modalities are observed, we approximate the true posterior $p(\mathbf{z} \mid \mathbf{x}_{\mathrm{o}}, \mathbf{x}_{\mathrm{m}}, \mathbf{y})$ with $q_\phi(\mathbf{z} \mid \mathbf{x}_{\mathrm{o}}, \mathbf{x}_{\mathrm{m}}, \mathbf{y})$ as the variational posterior. We use a conditional prior $p_\omega(\mathbf{z} \mid \mathbf{x}_{\mathrm{o}}, \mathbf{x}_{\mathrm{m}})$. We maximize the following ELBO:

$$\mathcal{L}_{\mathrm{complete}}^{\mathrm{ELBO}} = \mathbb{E}_{\mathbf{z} \sim q_\phi(\mathbf{z}|\mathbf{x}_{\mathrm{o}}, \mathbf{x}_{\mathrm{m}}, \mathbf{y})} \Big[ \log p_\theta(\mathbf{y} \mid \mathbf{x}_{\mathrm{o}}, \mathbf{z}) \Big] \\ - \mathrm{KL}(q_\phi(\mathbf{z} \mid \mathbf{x}_{\mathrm{o}}, \mathbf{x}_{\mathrm{m}}, \mathbf{y}) \,\|\, p_\omega(\mathbf{z} \mid \mathbf{x}_{\mathrm{o}}, \mathbf{x}_{\mathrm{m}})) . \tag{2}$$

**Case 2: Missing modality** ($\mathcal{D}_{\mathrm{missing}}$). When $\mathbf{x}_{\mathrm{m}}$ is missing, we use the variational posterior $q_\phi(\mathbf{z} \mid \mathbf{x}_{\mathrm{o}}, \mathbf{y})$ and the conditional prior $p_\omega(\mathbf{z} \mid \mathbf{x}_{\mathrm{o}})$. The resulting ELBO is

$$\mathcal{L}_{\mathrm{missing}}^{\mathrm{ELBO}} = \mathbb{E}_{\mathbf{z} \sim q_\phi(\mathbf{z}|\mathbf{x}_{\mathrm{o}}, \mathbf{y})} \Big[ \log p_\theta(\mathbf{y} \mid \mathbf{x}_{\mathrm{o}}, \mathbf{z}) \Big] \\ - \mathrm{KL}(q_\phi(\mathbf{z} \mid \mathbf{x}_{\mathrm{o}}, \mathbf{y}) \,\|\, p_\omega(\mathbf{z} \mid \mathbf{x}_{\mathrm{o}})) . \tag{3}$$

We jointly maximize both ELBOs across the training set to learn a shared latent representation in complete and missing-modality scenarios. Both ELBOs maximize $\log p_\theta(\mathbf{y} \mid \mathbf{x}_{\mathrm{o}}, \mathbf{z})$ and contain no reconstruction term for the missing modality.

When trained jointly, the unimodal and multimodal conditional priors can shift together in $\mathbf{z}$ without changing the KL. Because KL divergence is translation invariant, a common shift of both distributions leaves the KL invariant, creating a shift symmetry in $\mathbf{z}$. We break this symmetry by anchoring $p_\omega(\mathbf{z} \mid \mathbf{x}_{\mathrm{o}})$ to $\mathcal{N}(\mathbf{0}, \mathbf{I})$ (Mansimov et al., 2019) and tying $p_\omega(\mathbf{z} \mid \mathbf{x}_{\mathrm{o}}, \mathbf{x}_{\mathrm{m}})$ to $p_\omega(\mathbf{z} \mid \mathbf{x}_{\mathrm{o}})$ for the same $\mathbf{x}_{\mathrm{o}}$ using a regularizer $\mathcal{R}$:

$$\mathcal{R} = \sum_{i=1}^{N_c+N_m} \mathrm{KL}(p_\omega(\mathbf{z} \mid \mathbf{x}_{\mathrm{o},i}) \,\|\, \mathcal{N}(\mathbf{0}, \mathbf{I})) \\ + \sum_{i=1}^{N_c} \mathrm{KL}(p_\omega(\mathbf{z} \mid \mathbf{x}_{\mathrm{o},i}, \mathbf{x}_{\mathrm{m},i}) \,\|\, p_\omega(\mathbf{z} \mid \mathbf{x}_{\mathrm{o},i})) , \tag{4}$$

We parameterize the conditional priors $p_\omega(\mathbf{z} \mid \cdot)$ and the variational posteriors $q_\phi(\mathbf{z} \mid \cdot)$ as diagonal Gaussians, where the mean and variance are given by shared amortized networks. The conditioning variables depend on modality availability, with the priors conditioned on $\mathbf{x}_{\mathrm{o}}$ or $(\mathbf{x}_{\mathrm{o}}, \mathbf{x}_{\mathrm{m}})$, and the posteriors conditioned on $(\mathbf{x}_{\mathrm{o}}, \mathbf{y})$ or $(\mathbf{x}_{\mathrm{o}}, \mathbf{x}_{\mathrm{m}}, \mathbf{y})$.

$$p_\omega(\mathbf{z} \mid \cdot) = \mathcal{N}(\mathbf{z}; \mu_\omega(\cdot), \mathrm{diag}(\sigma_\omega(\cdot)^2)) , \\ q_\phi(\mathbf{z} \mid \cdot) = \mathcal{N}(\mathbf{z}; \mu_\phi(\cdot), \mathrm{diag}(\sigma_\phi(\cdot)^2)) . \tag{5}$$

To prevent posterior collapse, we follow Zhu et al. (2020) and apply batch normalization (BN) to the posterior mean $\mu_\phi(\cdot)$ with fixed scale $\gamma$ and learnable offset $\beta$. We compute BN statistics using mini-batch statistics during training. This prevents the posterior from trivially matching the prior by encouraging the KL term to remain non-zero. During training, we use the reparameterization trick to allow backpropagation through samples from these distributions:

$$\mathbf{z} = \mu_\phi(\cdot) + \sigma_\phi(\cdot) \odot \boldsymbol{\varepsilon}, \qquad \boldsymbol{\varepsilon} \sim \mathcal{N}(\mathbf{0}, \mathbf{I}), \tag{6}$$

where $\odot$ denotes the element-wise (Hadamard) product. The final training objective is

$$\max_{\theta, \phi, \omega} \sum_{i=1}^{N_c} \mathcal{L}_{\mathrm{complete},i}^{\mathrm{ELBO}} + \sum_{j=1}^{N_m} \mathcal{L}_{\mathrm{missing},j}^{\mathrm{ELBO}} - \mathcal{R}, \tag{7}$$

where we optimize jointly over all model parameters.

## 2.2. Inference

During testing, the labels $\mathbf{y}$ are unknown. We obtain predictions by marginalizing out the latent variable under the appropriate conditional prior and approximate the resulting integral via Monte Carlo sampling. We draw $K$ latent samples from the prior and average the resulting predictive probabilities:

$$p_\theta(\mathbf{y} \mid \mathbf{x}_o) \approx \frac{1}{K} \sum_{k=1}^{K} p_\theta(\mathbf{y} \mid \mathbf{x}_o, \mathbf{z}^{(k)}), \quad (8)$$
$$\mathbf{z}^{(k)} \sim p_\omega(\mathbf{z} \mid \mathbf{x}_o).$$

Following Figure 2, when both modalities are available at test time, we use the complete prior $p_\omega(\mathbf{z} \mid \mathbf{x}_o, \mathbf{x}_m)$.

To evaluate whether the missing modality is informative, we measure how the predictions change as we vary latent samples. We define $\mathcal{V} \equiv \mathcal{V}_{\mathbf{z}}[p_\theta(\cdot \mid \mathbf{x}_o, \mathbf{z})]$ as the expected total variation distance (TVD) between the predictive distribution $p_\theta(\cdot \mid \mathbf{x}_o, \mathbf{z})$ and its mean given by $\bar{p}_\theta(\cdot \mid \mathbf{x}_o) = \mathbb{E}_{\mathbf{z} \sim p_\omega(\mathbf{z} \mid \mathbf{x}_o)}[p_\theta(\cdot \mid \mathbf{x}_o, \mathbf{z})]$:

$$\mathcal{V} = \mathbb{E}_{\mathbf{z} \sim p_\omega(\mathbf{z} \mid \mathbf{x}_o)} \Big[ \mathrm{TVD}(p_\theta(\cdot \mid \mathbf{x}_o, \mathbf{z}), \bar{p}_\theta(\cdot \mid \mathbf{x}_o)) \Big]. \quad (9)$$

We denote this quantity by $\mathcal{V}_{\mathrm{missing}}$ when $\mathbf{z} \sim p_\omega(\mathbf{z} \mid \mathbf{x}_o)$, and by $\mathcal{V}_{\mathrm{complete}}$ when $\mathbf{z} \sim p_\omega(\mathbf{z} \mid \mathbf{x}_o, \mathbf{x}_m)$. Larger $\mathcal{V}_{\mathrm{missing}}$ indicates that $\mathbf{x}_m$ can substantially alter the predictions.

To characterize plausible outputs under missingness, we draw $\mathbf{z} \sim p_\omega(\mathbf{z} \mid \mathbf{x}_o)$, get the corresponding output logits from $p_\theta(\cdot \mid \mathbf{x}_o, \mathbf{z})$, and cluster these logits using a Dirichlet Process Gaussian Mixture Model (DPGMM). We label each cluster by its mean predicted class distribution, yielding a set of plausible labels for the input. If the clusters contain multiple plausible labels, this indicates that the latent variable (and thus the missing modality) significantly influences the prediction. Conversely, if the clusters are dominated by a single label, this suggests that the observed modality is sufficient.

## 3. Related Work

**Data imputation.** Missing data imputation has been studied extensively outside multimodal learning. Earlier works used simple heuristics such as zero-filling (Liu et al., 2023; Parthasarathy & Sundaram, 2020) and averaging-based variants such as mean/mode imputation or nearest neighbors. Many benchmarks evaluate imputation methods under different datasets and missingness assumptions (Luengo et al., 2012; Poulos & Valle, 2018; Woźnica & Biecek, 2020; Le Morvan et al., 2021; Shadbahr et al., 2023; Li et al., 2024; Morvan & Varoquaux, 2025). These evaluations focus on imputation quality rather than downstream predictive performance. Prior works have shown that improved imputations do not always translate to better downstream accuracy (Shadbahr et al., 2023; Morvan & Varoquaux, 2025). Under the missing completely at random assumption, Paterakis et al. (2024) similarly reports limited gains beyond simple mean and mode baselines. Similarly to our work, Ramchandran et al. (2024) proposes a latent-variable model that treats missing covariates as latent variables and marginalizes them out during inference. Their model, however, differs from ours in that it creates a single latent variable for all covariates, and their analysis does not investigate how the imputation distribution affects the predictive distribution in a fine-grained manner.

**Multimodal learning with missing modalities.** Many Variational Autoencoder (VAE)-based multimodal models have also been proposed to handle missing modalities (Suzuki et al., 2017; Vedantam et al., 2018; Tsai et al., 2019; Shi et al., 2019; Sutter et al., 2021; Gong et al., 2021; Joy et al., 2022; Palumbo et al., 2023). These methods focus on generative modeling by optimizing a marginal-likelihood objective via an ELBO, learning to reconstruct the inputs while regularizing the latent distribution toward a prior. As a result, $\mathbf{z}$ captures variation in the inputs, but it does not align $\mathbf{z}$ with the discriminative decision boundary required for modeling $p(\mathbf{y} \mid \cdot)$ under missing modalities. CMMD (Mancisidor et al., 2024) takes a step in this direction by incorporating a discriminative component into the multimodal latent framework, but assumes fully observed data during training. MEME (Joy et al., 2022) and VS-VAE (Gong et al., 2021) consider partial modality availability, where only a subset of training examples contains all modalities, but they also focus on generative modeling. In contrast, PRIMO focuses on discriminative prediction under heterogeneous modality availability during both training and inference. Additional details of these methods are provided in Section B of the Appendix.

**Multimodal learning with complete modalities.** One aspect of multimodal learning that has gained interest in recent years is the propensity of multimodal models to rely on a single modality rather than utilizing all available modalities (Agrawal et al., 2018; Singh et al., 2019; Dancette et al., 2021; Si et al., 2021; Madaan et al., 2024). More recently, the community has thus shifted its attention to analyzing these multimodal datasets by using various diagnostic checks and metrics. These include measuring performance change under modality removal or shuffling (Gu et al., 2025; Madaan et al., 2026), defining modality importance scores (Gat et al., 2021; Park et al., 2025), or circular evaluation (Liu et al., 2024). These approaches often lack either a way to analyze these multimodal data at an instance level or a mathematically interpretable justification. Our approach, on the other hand, allows us to inspect the impact of a (missing) modality at the level of individual instances in a fine-grained manner.

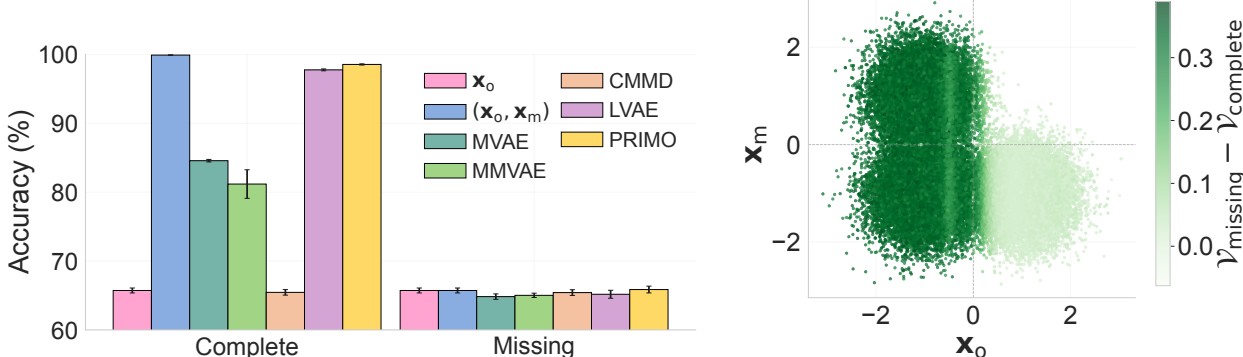

*Figure 3.* **Evaluation on the XOR dataset. (Left.)** Accuracy under complete and missing-modality inputs. PRIMO matches the unimodal baseline ($\mathbf{x}_o$) when $\mathbf{x}_m$ is missing and matches the multimodal baseline ($\mathbf{x}_o, \mathbf{x}_m$) when both modalities are observed, outperforming the remaining baselines. **(Right.)** Scatter plot of the predictive impact gap $\mathcal{V}_{\text{missing}} - \mathcal{V}_{\text{complete}}$. The gap is small for examples with $\mathbf{x}_o > 0$, where the label can be determined by $\mathbf{x}_o$ only, and larger for $\mathbf{x}_o < 0$, where $\mathbf{x}_m$ affects the label.

# 4. Experiments

We evaluate the effectiveness of PRIMO on a diverse set of multimodal datasets spanning synthetic, vision-audio, and healthcare settings. We use a synthetic XOR dataset, Audio-Vision MNIST (Liang et al., 2021) with missing audio or vision, and MIMIC-III (Johnson et al., 2016; Liang et al., 2021) with patient demographics (static) and clinical measurements (time-series). Dataset, hyperparameters, and architecture details are in Section C of the Appendix.

Across all datasets, we compare PRIMO under both complete and missing modality conditions against (i) a unimodal baseline that observes only $\mathbf{x}_o$, and (ii) a multimodal ($\mathbf{x}_o, \mathbf{x}_m$) baseline that observes both modalities when available. To evaluate when a missing modality is informative, we use our proposed metric $\mathcal{V}$ and the clustering analysis defined in Section 2.2. We report empirical cumulative distribution function (ECDF) of $\mathcal{V}$ over the test set to summarize its instance-level distribution.

## 4.1. Synthetic XOR

We consider two 1D modalities $\mathbf{x}_o$ and $\mathbf{x}_m$, always observing $\mathbf{x}_o$ and masking $\mathbf{x}_m$ at random with probability $0.5$. We sample ($\mathbf{x}_o, \mathbf{x}_m$) from a mixture of three Gaussians with $\sigma = 0.5$ centered at $(-1, -1)$, $(-1, 1)$, and $(1, -1)$, and assign XOR labels based on the signs of ($\mathbf{x}_o, \mathbf{x}_m$). This yields examples where for $\mathbf{x}_o < 0$ the label depends on $\mathbf{x}_m$, while for $\mathbf{x}_o > 0$ it can be determine by $\mathbf{x}_o$ only.

**Results.** Figure 3 (left) shows accuracy in complete and missing scenarios. Alongside unimodal and multimodal baselines, we compare against MVAE (Wu & Goodman, 2018) and MMVAE (Shi et al., 2019) (generative baselines), CMMD (Mancisidor et al., 2024) (discriminative missing-modality baseline), and LVAE (Ramchandran et al., 2024) (imputation for missing covariates).

With $\mathbf{x}_m$ missing, all methods perform comparably to the unimodal baseline using only $\mathbf{x}_o$. With complete inputs, only PRIMO and LVAE match the multimodal model evaluated on complete inputs, consistent with MVAE/MMVAE not being optimized for classification. In our setup, CMMD is not directly applicable to the complete-modality scenario because during inference it always uses the conditional prior $p_\omega(\mathbf{z} \mid \mathbf{x}_o)$, even when $\mathbf{x}_m$ is observed.

Figure 3 (right) shows the predictive impact gap between missing and complete scenarios, $\mathcal{V}_{\text{missing}} - \mathcal{V}_{\text{complete}}$. Examples on the left exhibit a larger gap because the label depends on both modalities, whereas examples on the right are predictable from $\mathbf{x}_o$ alone. This demonstrates that PRIMO captures the predictive impact of the missing modality. Section C in the Appendix further visualizes the latent space and predictions across methods.

## 4.2. Audio-vision MNIST (AV-MNIST)

AV-MNIST is a multimodal digit classification dataset with ten classes using written digits from the MNIST dataset (Lecun et al., 1998) and human spoken digits from the Free Spoken Digit Dataset (FSDD) (Jackson et al., 2018). To control the task difficulty, the dataset variant introduced by Liang et al. (2021) varies the information content in each modality. For audio samples, it adds real-world environmental sounds from the ESC-50 dataset (Piczak, 2015). For image samples, it uses PCA-based energy reduction. We mask either the audio modality or the image modality independently with probability $0.5$. We further vary the missing rate in Figure 19.

**Results.** Table 1 shows the accuracy when audio or vision modality is missing. In both scenarios, PRIMO performs comparably to the unimodal ($\mathbf{x}_o$) and the multimodal I2M2 (Madaan et al., 2024) baselines. We show comparison with additional baselines in Figure 16.

*Table 1.* **Accuracy on AV-MNIST.** We consider audio-missing and vision-missing settings. PRIMO performs comparably to the unimodal baseline that uses the available modality and to the multimodal baseline in both scenarios. Subscripts denote the standard deviation over five runs.

|  | Audio | Vision |
| --- | --- | --- |
| $\mathbf{x_o}$ | $64.23_{0.17}$ | $40.36_{0.80}$ |
| PRIMO | $63.06_{0.72}$ | $37.58_{1.29}$ |
| $(\mathbf{x_o}, \mathbf{x_m})$ | $71.14_{0.42}$ | $71.32_{0.30}$ |
| PRIMO | $68.17_{1.42}$ | $68.27_{1.35}$ |

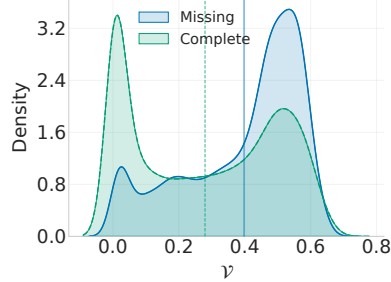

*Figure 4.* Distribution of $\mathcal{V}$ when audio is missing. Strong overlap between $\mathcal{V}_{\text{missing}}$ and $\mathcal{V}_{\text{complete}}$ indicates that predictions are often insensitive to the audio modality for those examples.

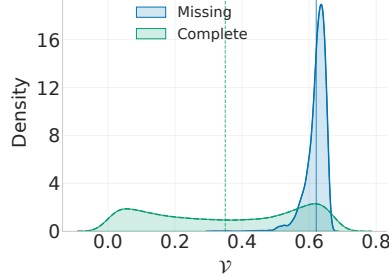

*Figure 5.* Distribution of $\mathcal{V}$ when vision is missing. $\mathcal{V}_{\text{missing}}$ is shifted to the right relative to $\mathcal{V}_{\text{complete}}$, indicating greater sensitivity to plausible vision completions.

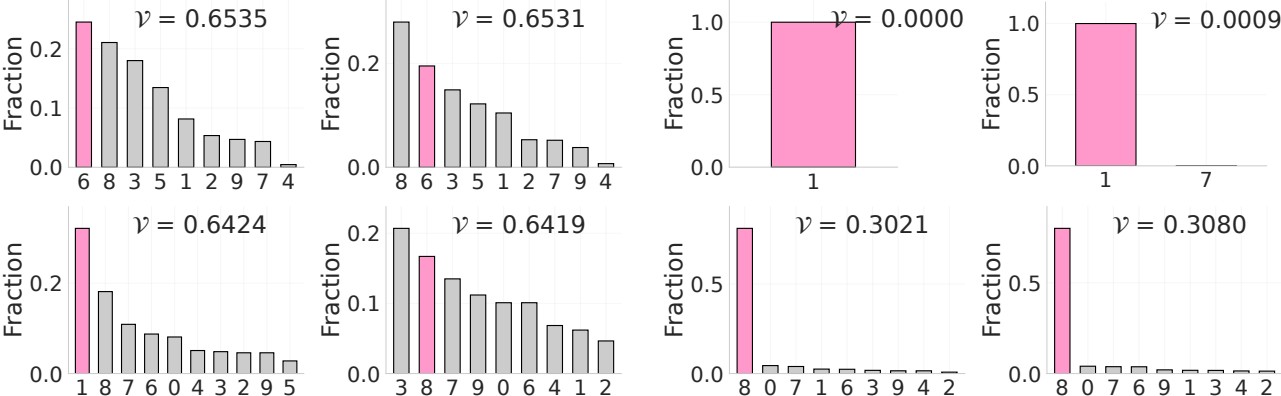

*Figure 6.* **Qualitative analysis of modality impact on AV-MNIST under audio-missing (top) and vision-missing (bottom).** We visualize plausible label outcomes induced by varying the latent completion $\mathbf{z}$. High-$\mathcal{V}$ examples yield multiple plausible label clusters under missingness, while low-$\mathcal{V}$ examples concentrate on a single dominant label.

We compare the distribution of $\mathcal{V}$ in both missing-modality scenarios in Figure 4 and Figure 5. Missing vision results in a significantly higher $\mathcal{V}$ ($\mu_{\text{miss}} = 0.57$) than missing audio ($\mu_{\text{miss}} = 0.37$). In contrast, when audio is missing, many examples exhibit $\mathcal{V}$ comparable to the complete input setting, suggesting that the prediction is often insensitive to the audio modality for those instances.

In Figure 6, we further characterize how the latent $\mathbf{z}$ capturing the missing modality affects predictive distribution using the clustering analysis from Section 2.2. We visualize which labels are most likely under the missing-audio (top row) and missing-vision (bottom row) settings. We visualize high- and low-variance examples with their corresponding $\mathcal{V}_{\text{missing}}$ reported as $\mathcal{V}$. We observe that high-$\mathcal{V}$ examples often correspond to multiple plausible labels, reflecting that different latent completions of the missing modality can change the predicted label distribution. For low-$\mathcal{V}$ examples, there is a concentration on a single dominant label across different latent completions in both settings. These results illustrate that PRIMO captures how missing modalities alter the set of plausible predictions differently for different examples.

### 4.3. MIMIC-III

MIMIC-III (Johnson et al., 2016) is a clinical dataset that contains Electronic Health Records (EHR) data from approximately 40,000 patients at Beth Israel Deaconess Medical Center from 2001 to 2012. We use two modalities: (a) static modality containing patient-level information such as age, admission type, and chronic conditions (acquired immunodeficiency syndrome, hematologic malignancy, and metastatic cancer), and (b) time-series modality with 12 physiological measurements recorded hourly over the first 24 hours after ICU admission. We use the processed benchmark version (Purushotham et al., 2018; Liang et al., 2021), which applies forward and backward filling, with mean imputation for time-series features that are entirely missing. We investigate the predictive impact of the time-series modality across multiple tasks by masking it at random with probability 0.5 and vary the missing rate in Figure 20. We consider mortality prediction as a 6-class problem (death within 1 day, 2 days, 3 days, 1 week, 1 year, or >1 year) and two ICD-9 groups binary prediction tasks, where we consider Group 1 (codes 140–239, neoplasms) and Group 7 (codes 460–519, respiratory diseases).

*Table 2.* **MIMIC-III accuracy.** We consider mortality and ICD-9 group prediction under missing and complete modality settings. We report mean and standard deviation across five runs for the unimodal baseline $(\mathbf{x}_o)$, the multimodal baseline $(\mathbf{x}_o, \mathbf{x}_m)$, and PRIMO in each setting.

| | Mortality | ICD-9 | |
| --- | --- | --- | --- |
| | | 140–239 | 460–519 |
| $\mathbf{x}_o$ | $76.36_{0.01}$ | $91.42_{0.00}$ | $56.22_{0.46}$ |
| PRIMO | $76.17_{0.07}$ | $91.41_{0.01}$ | $54.95_{1.44}$ |
| $(\mathbf{x}_o, \mathbf{x}_m)$ | $77.89_{0.17}$ | $91.37_{0.10}$ | $68.22_{0.52}$ |
| PRIMO | $77.08_{0.25}$ | $91.41_{0.03}$ | $65.78_{1.08}$ |

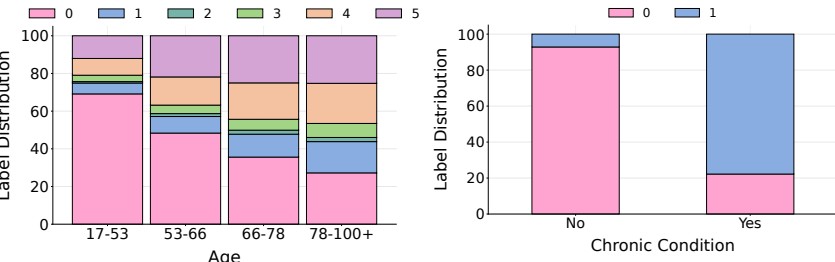

*Figure 7.* **(Left)** Cluster-induced plausible label distribution for mortality prediction stratified by age. **(Right)** Cluster-induced plausible label distribution for ICD-9 neoplasms (140–239) stratified by chronic condition. Distributions are computed from predictions across latent completions of the time-series modality.

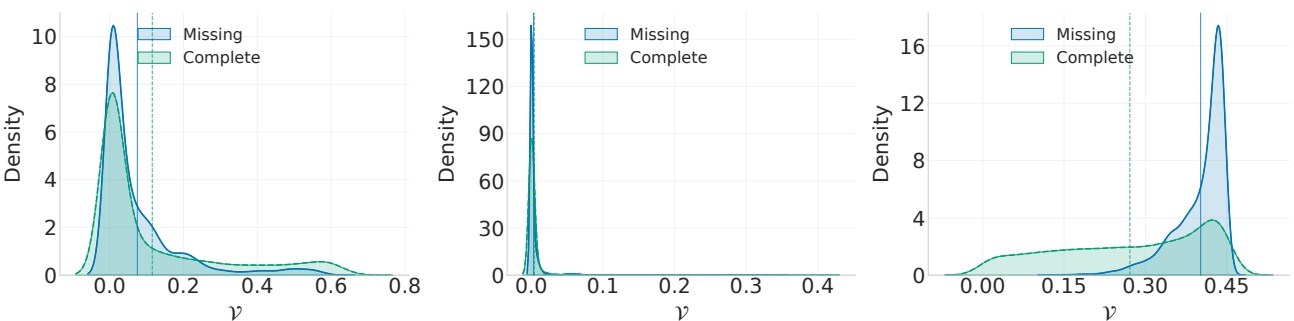

*Figure 8.* **Predictive impact under missing and complete time-series modality with sampling z.** We compare $\mathcal{V}$ for **(left)** mortality prediction, **(center)** ICD-9 140–239 (neoplasms), and **(right)** ICD-9 460–519 (respiratory diseases). The time-series modality has little impact for ICD-9 140–239, but it affects ICD-9 460–519 and mortality prediction.

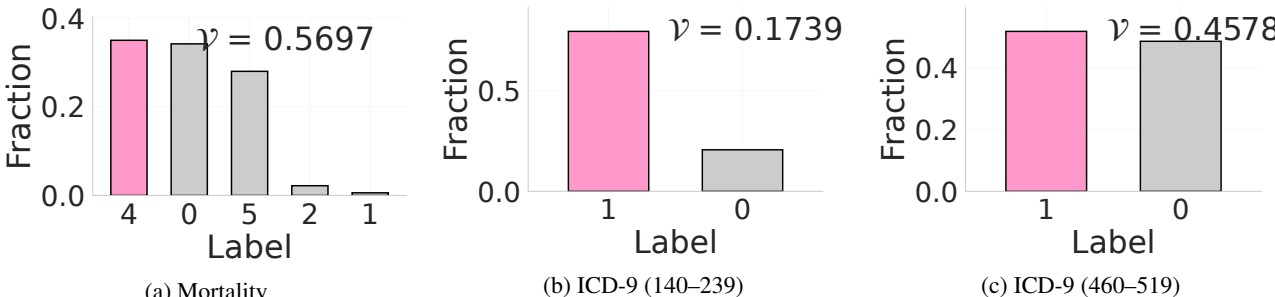

| (a) Mortality | (b) ICD-9 (140–239) | (c) ICD-9 (460–519) |
| --- | --- | --- |

*Figure 9.* **Patient-level analysis on MIMIC-III under missing time-series modality.** We report $\mathcal{V}_{missing}$ as $\mathcal{V}$ and visualize the fraction of clusters assigned to each label. High-$\mathcal{V}$ examples yield clusters spread across multiple labels, indicating ambiguity for (a) mortality risk and (c) respiratory disease diagnosis, whereas (b) neoplasm prediction concentrates on a single label and remains stable.

**Mortality prediction.** Time-series modality is often assumed to be important (Liang et al., 2021; Madaan et al., 2024) for this task since patient trajectories can capture deterioration patterns beyond static features. Table 2 shows that the aggregate performance gain from including the time-series modality is relatively small. $\mathcal{V}$ in Figure 8 shows that for most patients, the predictions are stable across plausible completions of the time series modality.

To investigate the tail of patients in this distribution, we conduct the clustering-based analysis from Section 2.2. Figure 7 (left) shows that the resulting label distribution from the clusters shifts towards high-risk mortality outcomes as age increases. This suggests that time-series might be more informative for older-aged patients. We show plausible labels for individual patients in Figure 9

(in the main text) and Figure 13 (in the Appendix). Consistent with our results, we observe that low-$\mathcal{V}$ cases usually correspond to low-risk predictions. In contrast, high-$\mathcal{V}$ cases concentrate among patients closer to high-risk mortality classes, where time-series modality can be critical.

**ICD-9 code prediction.** For predicting neoplasms (ICD-9 140–239), we obtain high accuracy in Table 2 and Figure 17 and low $\mathcal{V}_{missing}$ in Figure 8 despite the absence of time-series modality. This suggests that static modality is sufficient for this task. This is consistent with our clustering analysis in Figure 9 and Figure 14, where the predictions do not change much even in high-$\mathcal{V}$ samples and are dominated by a single label. This is supported by static modality containing chronic disease features, which are informative descriptors for this ICD block (see Figure 7 (right)).

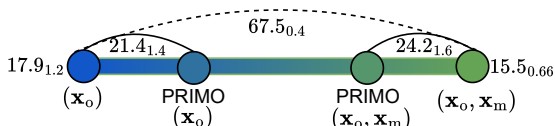

*Figure 10.* **Bias analysis with vision missing.** $\mathrm{PRIMO}(\mathbf{x}_\mathrm{o})$ stays close to the unimodal oracle, while $\mathrm{PRIMO}(\mathbf{x}_\mathrm{o}, \mathbf{x}_\mathrm{m})$ stays close to the multimodal oracle. The dashed arc shows the unimodal–multimodal oracle gap.

In contrast, for respiratory disease (ICD-9 460-519), we obtain near-random performance in Table 2, Figure 17 and high $\mathcal{V}_\mathrm{missing}$ when the time-series modality is missing in Figure 8. This is because respiratory diagnoses in the ICU depends on various time-series measurements. For instance, oxygenation-related variables such as $\mathrm{PaO_2/FiO_2}$ are direct indicators of respiratory impairment, other features such as temperature, WBC count, heart rate capture systemic instability, and infection that often co-occur with respiratory disease. Our patient-level analysis in Figure 9 and Figure 15 shows that missing time series leads to high-$\mathcal{V}_\mathrm{missing}$ and ambiguity in the predictions for most examples.

Overall, for a given dataset, depending on the task, modality importance can vary significantly. While for neoplasms, time series is often not essential, it plays an important role for respiratory diseases. These results show that PRIMO captures these nuances, obtaining good performance while providing patient-level analysis.

### 4.4. Bias Analysis

Let $p^*(\cdot \mid \mathbf{x}_\mathrm{o}) = \mathbb{E}[\mathbf{y} \mid \mathbf{x}_\mathrm{o}]$ denote the Bayes-optimal predictor given only $\mathbf{x}_\mathrm{o}$. PRIMO defines a predictive distribution conditioned on $(\mathbf{x}_\mathrm{o}, \mathbf{z})$, $p_\theta(\cdot \mid \mathbf{x}_\mathrm{o}, \mathbf{z})$. Marginalizing over the learned conditional prior gives

$$\bar{p}_\theta(\cdot \mid \mathbf{x}_\mathrm{o}) = \mathbb{E}_{\mathbf{z} \sim p_\omega(\mathbf{z}|\mathbf{x}_\mathrm{o})}\big[p_\theta(\cdot \mid \mathbf{x}_\mathrm{o}, \mathbf{z})\big]. \qquad (10)$$

We measure the discrepancy between this mean prediction and the Bayes-optimal unimodal predictor using TVD:

$$\mathcal{B}_\mathrm{missing} = \mathrm{TVD}(p^*(\cdot \mid \mathbf{x}_\mathrm{o}), \, \bar{p}_\theta(\cdot \mid \mathbf{x}_\mathrm{o})). \qquad (11)$$

When both modalities $(\mathbf{x}_\mathrm{o}, \mathbf{x}_\mathrm{m})$ are observed, we define $\bar{p}_\theta(\cdot \mid \mathbf{x}_\mathrm{o}, \mathbf{x}_\mathrm{m})$ analogously by replacing the prior with $p_\omega(\mathbf{z} \mid \mathbf{x}_\mathrm{o}, \mathbf{x}_\mathrm{m})$, and we define $\mathcal{B}_\mathrm{complete}$ by comparing to the Bayes-optimal multimodal predictor $p^*(\cdot \mid \mathbf{x}_\mathrm{o}, \mathbf{x}_\mathrm{m}) = \mathbb{E}[\mathbf{y} \mid \mathbf{x}_\mathrm{o}, \mathbf{x}_\mathrm{m}]$. This quantifies how well the learned priors recover the Bayes-optimal unimodal and multimodal predictors after marginalizing over $\mathbf{z}$.

To obtain an unbiased estimate, we partition the dataset into two disjoint halves. Using the complete-modality half, we train unimodal and multimodal oracles. Using the remaining half, we train PRIMO with a $50\%$ missing rate and evaluate it under the same missingness pattern at inference time for both missing and complete scenarios.

Figure 10 shows the bias for the missing-vision setting. The oracle distances provide a practical lower bound and are non-zero due to finite-sample effects and optimization noise. Under missing vision modality, PRIMO $(\mathbf{x}_\mathrm{o})$ is closer to the unimodal oracle trained on $\mathbf{x}_\mathrm{o}$. When both modalities are available, PRIMO $(\mathbf{x}_\mathrm{o}, \mathbf{x}_\mathrm{m})$ is closer to the multimodal oracle trained on $(\mathbf{x}_\mathrm{o}, \mathbf{x}_\mathrm{m})$, consistent with the unimodal–multimodal oracle gap induced by observing $\mathbf{x}_\mathrm{m}$.

## 5. Limitations and Future Work

**Constraints on validating modality importance.** In practical multimodal settings, we often do not have access to certain modalities at inference time, and in many applications labels may also be missing. This makes it challenging to evaluate whether instance-level modality impact estimates are correct. Our qualitative results on MIMIC-III suggest that modality relevance can vary substantially across tasks and examples within the same dataset. Validating this instance-level modality importance without any ground truth, however, remains an open problem. Incorporating human feedback with automated evaluation protocols would be an interesting direction for future work.

**Evaluation with many modalities.** Another practical constraint in multimodal learning is the scarcity of benchmarks with multiple modalities and heterogeneous missingness patterns. PRIMO can extend to any number of modalities by introducing a latent variable for each potentially missing modality, however, current standard benchmarks focus primarily on audio, vision, and text. This limits evaluation in settings with sensory data, tabular data, and multiple imaging modalities. We hope this work motivates benchmarks with three or more modalities and heterogeneous missingness patterns, enabling more realistic evaluation of imputation-based multimodal learning and instance-level modality importance under incomplete data.

## 6. Conclusion

We propose PRIMO, a supervised latent-variable model for characterizing predictions under plausible completions of a missing modality. PRIMO supports both complete and missing-modality settings, and achieves performance comparable to unimodal baselines when a modality is missing and multimodal baselines when all modalities are observed. Beyond predictive performance, PRIMO provides instance-level estimates of how missing modalities affect predictions across datasets and modality combinations. We find that modality contributions vary across tasks and across examples within the same dataset. These results highlight the heterogeneity of multimodal datasets. PRIMO provides a principled way to capture this heterogeneity as modality availability and relevance change.

## Impact Statement

Real-world data is often multimodal, but datasets rarely contain all modalities for every example. PRIMO enables multimodal learning under both complete and partial-modality settings by training and predicting with whatever modalities are available, while explicitly characterizing how missing modalities could affect predictions. In settings where acquiring an additional modality is expensive, slow, or risky, such as healthcare, PRIMO can support data acquisition. It can identify when additional measurements are likely to change the prediction for a particular patient and when they are unlikely to matter. This can enable more targeted acquisition policies. It can further reduce unnecessary collection and can lower costs while preserving predictive performance.

## Acknowledgement

This work was supported by the Institute of Information & Communications Technology Planning & Evaluation (IITP) with a grant funded by the Ministry of Science and ICT (MSIT) of the Republic of Korea in connection with the Global AI Frontier Lab International Collaborative Research, Samsung Advanced Institute of Technology (under the project Next Generation Deep Learning: From Pattern Recognition to AI), National Science Foundation (NSF) award No. 1922658, Center for Advanced Imaging Innovation and Research (CAI2R), National Center for Biomedical Imaging and Bioengineering operated by NYU Langone Health, and National Institute of Biomedical Imaging and Bioengineering through award number P41EB017183. The computational requirements for this work were supported by NYU IT High Performance Computing resources, services, and staff expertise and NYU Langone High Performance Computing Core's resources and personnel. This work was partly supported in part by the NYUAD Center for Interdisciplinary Data Science & AI (CIDSAI), funded by Tamkeen under the NYUAD Research Institute Award CG016. This content is solely the responsibility of the authors and does not represent the views of the funding agencies.

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

**Organization.** The Appendix includes ELBO derivations (Section A), detailed related work on multimodal learning with missing modalities (Section B), and the experimental setup with additional results (Section C).

## A. ELBO derivations

### A.1. Complete modalities ($\mathcal{D}_{\text{complete}}$)

When both modalities $\mathbf{x}_o$ and $\mathbf{x}_m$ are available, we maximize the conditional log-likelihood $\log p(\mathbf{y} \mid \mathbf{x}_o, \mathbf{x}_m)$. Following the dependencies from our graphical model in Figure 2, the joint distribution of the label and latent variable factorizes as $p(\mathbf{y}, \mathbf{z} \mid \mathbf{x}_o, \mathbf{x}_m) = p_\theta(\mathbf{y} \mid \mathbf{x}_o, \mathbf{z}) p_\omega(\mathbf{z} \mid \mathbf{x}_o, \mathbf{x}_m)$, where $\mathbf{y} \perp \mathbf{x}_m \mid \{\mathbf{z}, \mathbf{x}_o\}$. By introducing the variational posterior $q_\phi(\mathbf{z} \mid \mathbf{x}_o, \mathbf{x}_m, \mathbf{y})$, we derive the ELBO as follows:

$$
\begin{aligned}
\log p(\mathbf{y} \mid \mathbf{x}_o, \mathbf{x}_m) &= \log \int p_\theta(\mathbf{y} \mid \mathbf{x}_o, \mathbf{z}) p_\omega(\mathbf{z} \mid \mathbf{x}_o, \mathbf{x}_m) \, d\mathbf{z} \\
&= \log \int q_\phi(\mathbf{z} \mid \mathbf{x}_o, \mathbf{x}_m, \mathbf{y}) \frac{p_\theta(\mathbf{y} \mid \mathbf{x}_o, \mathbf{z}) p_\omega(\mathbf{z} \mid \mathbf{x}_o, \mathbf{x}_m)}{q_\phi(\mathbf{z} \mid \mathbf{x}_o, \mathbf{x}_m, \mathbf{y})} \, d\mathbf{z} \\
&\geq \int q_\phi(\mathbf{z} \mid \mathbf{x}_o, \mathbf{x}_m, \mathbf{y}) \log \left( \frac{p_\theta(\mathbf{y} \mid \mathbf{x}_o, \mathbf{z}) p_\omega(\mathbf{z} \mid \mathbf{x}_o, \mathbf{x}_m)}{q_\phi(\mathbf{z} \mid \mathbf{x}_o, \mathbf{x}_m, \mathbf{y})} \right) d\mathbf{z} \quad \text{(Jensen's Inequality)} \\
&= \mathbb{E}_{q_\phi} \left[ \log p_\theta(\mathbf{y} \mid \mathbf{x}_o, \mathbf{z}) \right] + \mathbb{E}_{q_\phi} \left[ \log \frac{p_\omega(\mathbf{z} \mid \mathbf{x}_o, \mathbf{x}_m)}{q_\phi(\mathbf{z} \mid \mathbf{x}_o, \mathbf{x}_m, \mathbf{y})} \right] \\
&= \mathbb{E}_{q_\phi(\mathbf{z} \mid \mathbf{x}_o, \mathbf{x}_m, \mathbf{y})} \left[ \log p_\theta(\mathbf{y} \mid \mathbf{x}_o, \mathbf{z}) \right] - \mathrm{KL} \left( q_\phi(\mathbf{z} \mid \mathbf{x}_o, \mathbf{x}_m, \mathbf{y}) \,\|\, p_\omega(\mathbf{z} \mid \mathbf{x}_o, \mathbf{x}_m) \right) .
\end{aligned}
$$

### A.2. Missing modality ($\mathcal{D}_{\text{missing}}$)

In the case where $\mathbf{x}_m$ is unavailable, we maximize $\log p(\mathbf{y} \mid \mathbf{x}_o)$. The dashed edge in Figure 2 indicates a correlation between $\mathbf{x}_o$ and $\mathbf{x}_m$, implying that $\mathbf{x}_o$ carries information about the missing modality. We thus utilize a conditional prior $p_\omega(\mathbf{z} \mid \mathbf{x}_o)$ to infer $\mathbf{z}$. Using the variational posterior $q_\phi(\mathbf{z} \mid \mathbf{x}_o, \mathbf{y})$, we derive the ELBO as:

$$
\begin{aligned}
\log p(\mathbf{y} \mid \mathbf{x}_o) &= \log \int p_\theta(\mathbf{y} \mid \mathbf{x}_o, \mathbf{z}) p_\omega(\mathbf{z} \mid \mathbf{x}_o) \, d\mathbf{z} \\
&= \log \int q_\phi(\mathbf{z} \mid \mathbf{x}_o, \mathbf{y}) \frac{p_\theta(\mathbf{y} \mid \mathbf{x}_o, \mathbf{z}) p_\omega(\mathbf{z} \mid \mathbf{x}_o)}{q_\phi(\mathbf{z} \mid \mathbf{x}_o, \mathbf{y})} \, d\mathbf{z} \\
&\geq \int q_\phi(\mathbf{z} \mid \mathbf{x}_o, \mathbf{y}) \log \left( \frac{p_\theta(\mathbf{y} \mid \mathbf{x}_o, \mathbf{z}) p_\omega(\mathbf{z} \mid \mathbf{x}_o)}{q_\phi(\mathbf{z} \mid \mathbf{x}_o, \mathbf{y})} \right) d\mathbf{z} \quad \text{(Jensen's Inequality)} \\
&= \mathbb{E}_{q_\phi} \left[ \log p_\theta(\mathbf{y} \mid \mathbf{x}_o, \mathbf{z}) \right] - \mathbb{E}_{q_\phi} \left[ \log \frac{q_\phi(\mathbf{z} \mid \mathbf{x}_o, \mathbf{y})}{p_\omega(\mathbf{z} \mid \mathbf{x}_o)} \right] \\
&= \mathbb{E}_{q_\phi(\mathbf{z} \mid \mathbf{x}_o, \mathbf{y})} \left[ \log p_\theta(\mathbf{y} \mid \mathbf{x}_o, \mathbf{z}) \right] - \mathrm{KL} \left( q_\phi(\mathbf{z} \mid \mathbf{x}_o, \mathbf{y}) \,\|\, p_\omega(\mathbf{z} \mid \mathbf{x}_o) \right) .
\end{aligned}
$$

## B. Related Work on Multimodal learning with Missing Modalities

Many prior multimodal VAE studies (Suzuki et al., 2017; Vedantam et al., 2018; Tsai et al., 2019; Shi et al., 2019; Sutter et al., 2021; Gong et al., 2021; Joy et al., 2022; Palumbo et al., 2023) focus on generative modeling by optimizing the marginal likelihood $\max_\theta \mathbb{E}_{q(\mathbf{z} \mid \mathbf{x}_o)} \left[ \log p_\theta(\mathbf{x}_o \mid \mathbf{z}) \right] - \beta \mathrm{KL}(q(\mathbf{z} \mid \mathbf{x}_o) \| p(\mathbf{z}))$ rather than improving discriminative performance under missing modalities. JMVAE (Suzuki et al., 2017) and tELBO (Vedantam et al., 2018) model the joint distribution $p(\mathbf{x}, \mathbf{x}')$. These methods use paired multimodal examples during training to learn an inference network $p_\theta(\mathbf{z} \mid \mathbf{x}, \mathbf{x}')$ conditioned on all modalities. To scale beyond two modalities, MVAE (Wu & Goodman, 2018) combines modality-specific posteriors using product-of-experts $q_\phi(\mathbf{z} \mid \mathbf{x}_{1:M}) \propto \prod_m q_{\phi_m}(\mathbf{z} \mid \mathbf{x}_m)$, while MMVAE (Shi et al., 2019) uses a mixture of experts $q_\phi(\mathbf{z} \mid \mathbf{x}_{1:M}) = \sum_m \pi_m q_{\phi_m}(\mathbf{z} \mid \mathbf{x}_m)$. MoPoE (Sutter et al., 2021) further generalizes these objectives using a mixture of product of experts. These methods commonly rely on sub-sampling modality subsets, which imposes an undesirable upper bound on the multimodal ELBO (Daunhawer et al., 2022). Similarly, while mmJSD (Sutter et al., 2020) and MMVAE+ (Palumbo et al., 2023) seperated modality-specific and shared latent spaces, they fundamentally optimize $\log p(\mathbf{x}_o)$ and use fully paired multimodal data for training.

Optimizing a generative ELBO learns a $\mathbf{z}$ that captures the input variation, which might not align with the optimal class decision boundary required for modeling $p(\mathbf{y} \mid \cdot)$. CMMD (Mancisidor et al., 2024) took a step in this direction by incorporating a discrimiantive component into the multimodal latent framework, but assumes fully observed multimodal data during training. Using fully paired multimodal data in this setup creates a train-test mismatch, the posterior learnt during training $q(\mathbf{z} \mid \mathbf{x}_o, \mathbf{x}_m, \mathbf{y})$ has access to $\mathbf{x}_m$ but at test time only $\mathbf{x}_o$ is available. MEME (Joy et al., 2022) and VSVAE (Gong et al., 2021) considered partial multimodal missignenss (only a subset of training examples contains all modalities), but they were limited to generative modeling. We argue that both these settings are crucial because we need to align the $\mathbf{z}$ when the modality is observed and missing. PRIMO explicitly addresses the above limitations and is designed for discriminative setups when modalities are partially observed during both training and testing.

## C. Additional Experiments

This section summarizes the hyperparameters and architectural details (see Section C.1) with additional experimental results (see Section C.2).

### C.1. Experimental Setup

**XOR.** We generate a synthetic 2D XOR classification task with 40,000 samples drawn from four Gaussian clusters centered at $(\pm 1, \pm 1)$ with standard deviation 0.5. Three quadrants are used to create a dataset where $\mathbf{x}_m$ provides non-redundant information for classification. We use a 70/30 train/test split and randomly drop the second modality $\mathbf{x}_m$ for 50% of training examples. Each modality is a single scalar feature encoded through a shared two-layer MLP architecture. The prior and posterior are two-layer MLPs with hidden dimension 128, projecting to a two-dimensional latent space. We train with AdamW (learning rate $1 \times 10^{-3}$ and weight decay $1 \times 10^{-4}$). For evaluation, we use 200 Monte Carlo samples. Results are averaged over four random seeds.

**AVMNIST.** We use modality-specific LeNet encoders (three layers for vision and five layers for audio), each followed by a linear projection to a 128-dimensional latent space. The prior and posterior are implemented as MLPs, with the posterior conditioned on both modalities and the label, and the prior conditioned on the fused representation. We share the same prior/posterior parameters across complete and missing-modality scenarios; when a modality is missing, its representation is zeroed out before fusion. We train with AdamW (learning rate $5 \times 10^{-4}$). At evaluation, results are estimated using 2000 Monte Carlo samples.

**MIMIC-III.** We use an $80/10/10$ train/validation/test split and randomly drop time-series modality for $50\%$ of training examples. Static features are encoded with a two-layer MLP and time-series features with a GRU; both are projected to a 64-dimensional latent space. The prior and posterior are MLPs conditioned on the available modalities (and the label for the posterior). As in AVMNIST, we share prior/posterior parameters across complete and missing-modality settings, and zero out the representation of any missing modality. We train with AdamW (learning rate $5 \times 10^{-4}$). For evaluation, we use 500 Monte Carlo samples.

### C.2. Additional Results

We report XOR predictions and latent-space visualizations in Figure 11 and Figure 12, respectively. We also provide additional qualitative examples for mortality prediction in Figure 13, ICD-9 140–239 (neoplasms) prediction in Figure 14, and ICD-9 460–519 (respiratory diseases) prediction in Figure 15. We compare the performance of PRIMO with baselines on AV-MNIST in Figure 16 and MIMIC-III in Figure 17.

We additionally evaluate the effect of the number of MC samples on accuracy in Figure 18, and the effect of training missing rate on test accuracy on AV-MNIST in Figure 19 and MIMIC-III in Figure 20.

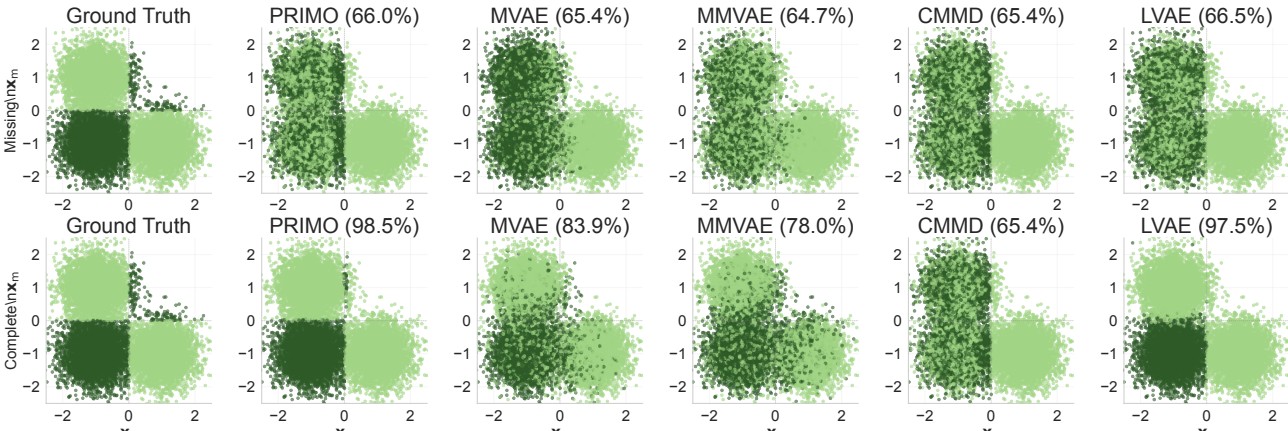

*Figure 11.* **XOR predictions under missing and complete modalities.** Each column shows a method and each row corresponds to the modality-availability setting (top: $x_m$ missing, bottom: complete). For each input $x_o$, we sample latent completions and visualize the induced distribution over predicted labels; points are colored by the predicted class. Methods that capture label-relevant uncertainty produce multiple plausible labels in regions where the XOR label depends on $x_m$ (e.g., $x_o < 0$), while predictions concentrate on a single label when $x_o$ is sufficient (e.g., $x_o > 0$). Accuracies are shown in the column headers.

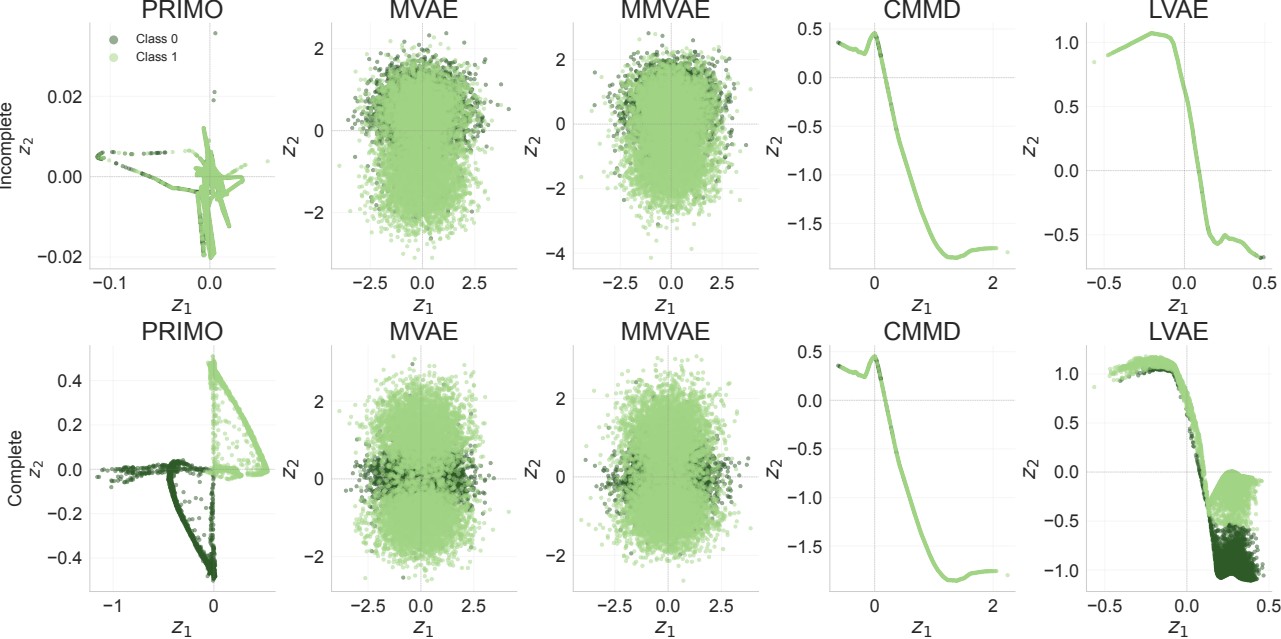

*Figure 12.* **XOR latent-space structure across methods.** We visualize the 2D latent representations used by each method for incomplete inputs (top row) and complete inputs (bottom row). Points are colored by the predicted class under the corresponding latent sample.

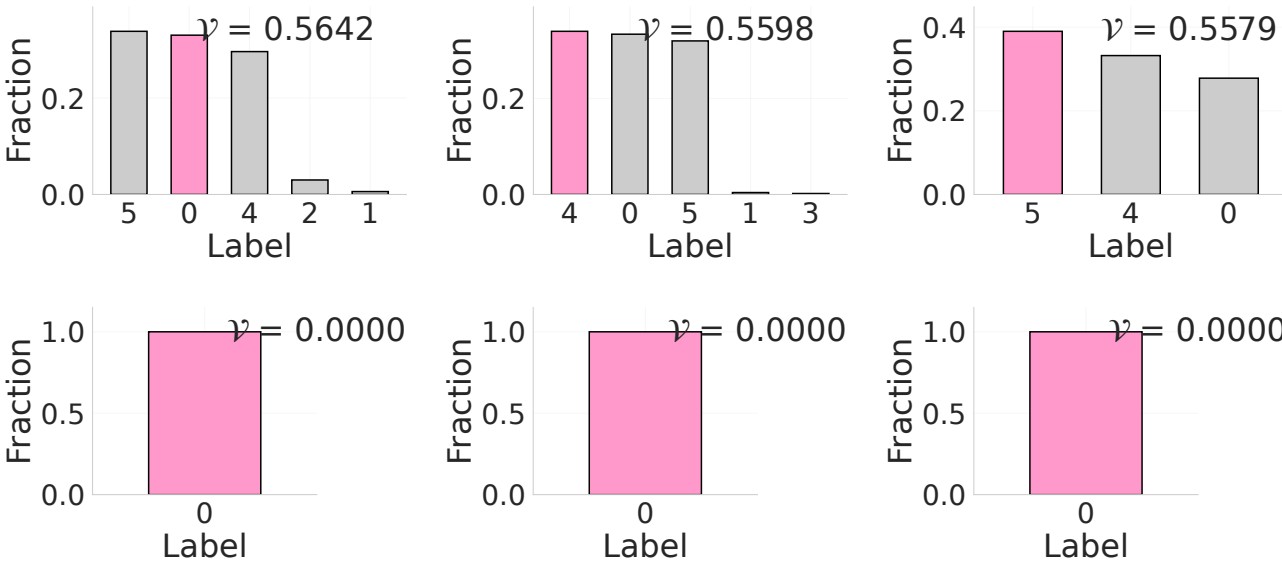

*Figure 13.* **Patient-level mortality results on MIMIC-III under missing time-series inputs.** Each panel shows the fraction of clusters assigned to each mortality class, with $\mathcal{V}_{\text{missing}}$ reported as $\mathcal{V}$. The top row shows high-$\mathcal{V}$ examples with clusters spread across multiple risk classes, while the bottom row shows low-$\mathcal{V}$ examples dominated by a single class.

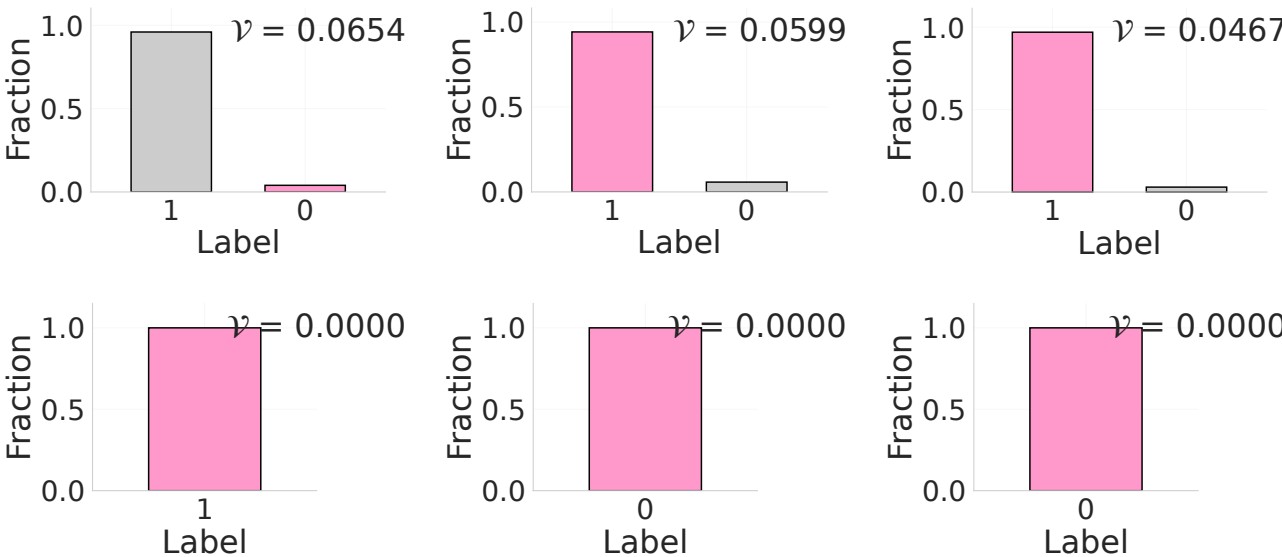

*Figure 14.* **Patient-level ICD-9 (140–239) results on MIMIC-III under missing time-series inputs.** Each panel shows the fraction of clusters assigned to each label, with $\mathcal{V}_{\text{missing}}$ reported as $\mathcal{V}$. The top row shows high-$\mathcal{V}$ examples and the bottom row shows low-$\mathcal{V}$ examples; in both cases, clusters are largely dominated by a single label, indicating stable predictions.

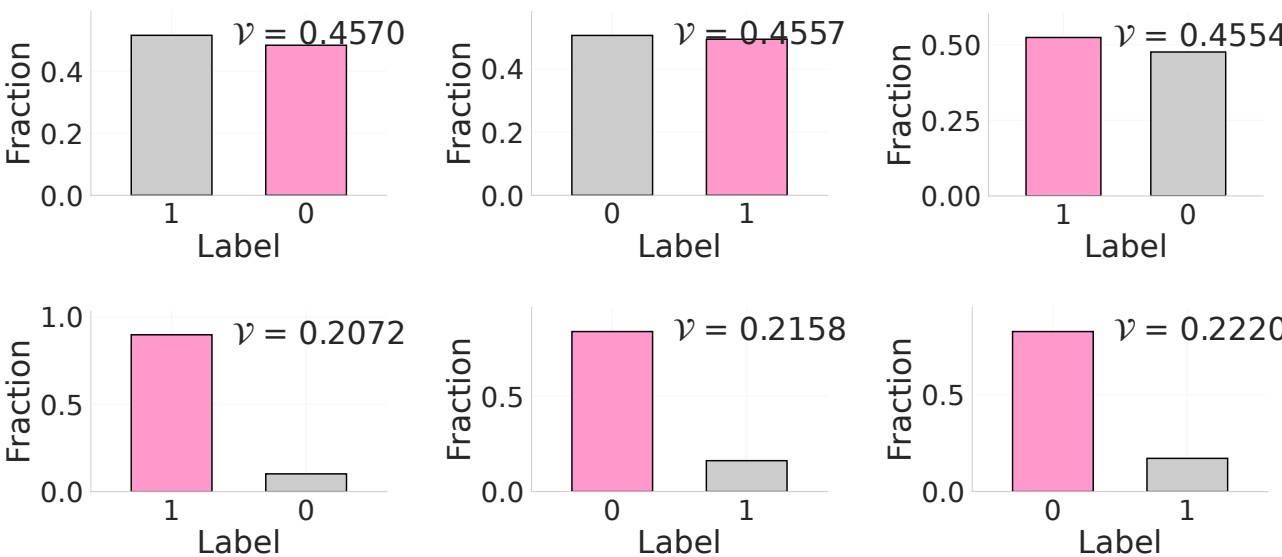

*Figure 15.* **Patient-level ICD-9 (460–519) results on MIMIC-III under missing time-series inputs.** Each panel shows the fraction of clusters assigned to each label, with $\mathcal{V}_{\text{missing}}$ reported as $\mathcal{V}$. The top row shows high-$\mathcal{V}$ examples with clusters spread across multiple labels, indicating ambiguity when time-series is missing, while the bottom row shows low-$\mathcal{V}$ examples dominated by a single label.

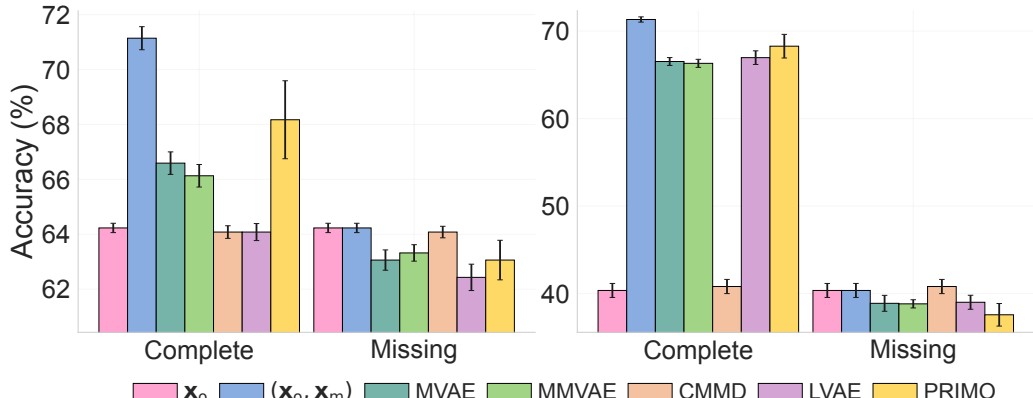

*Figure 16.* **Comparison with baselines on the AV-MNIST dataset.** Results with the audio modality missing (left) and the image modality missing (right). None of the baselines consistently match the performance of the oracle or PRIMO across both complete and missing-modality scenarios.

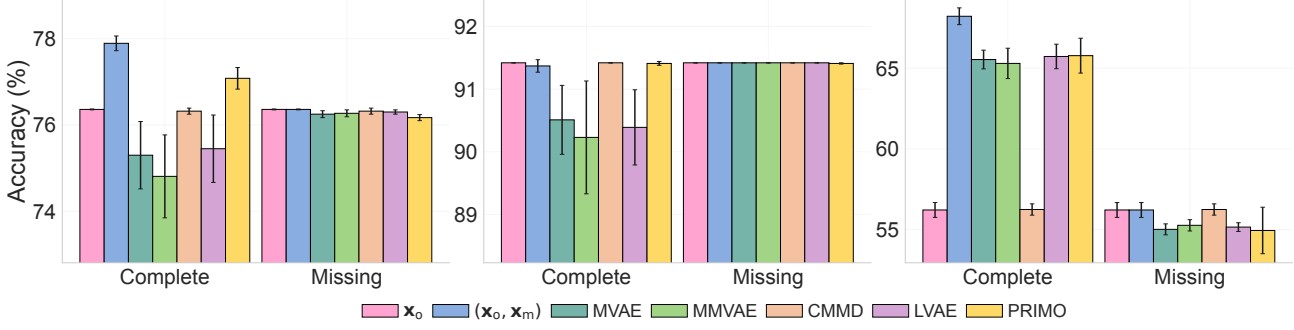

*Figure 17.* **Comparison with baselines on the MIMIC-III dataset.** Results on mortality prediction (left), ICD-9 code group prediction 140–239 (middle), and ICD-9 code group prediction 460–519 (right). None of the baselines consistently match the performance of the oracle or PRIMO across both complete and missing-modality scenarios.

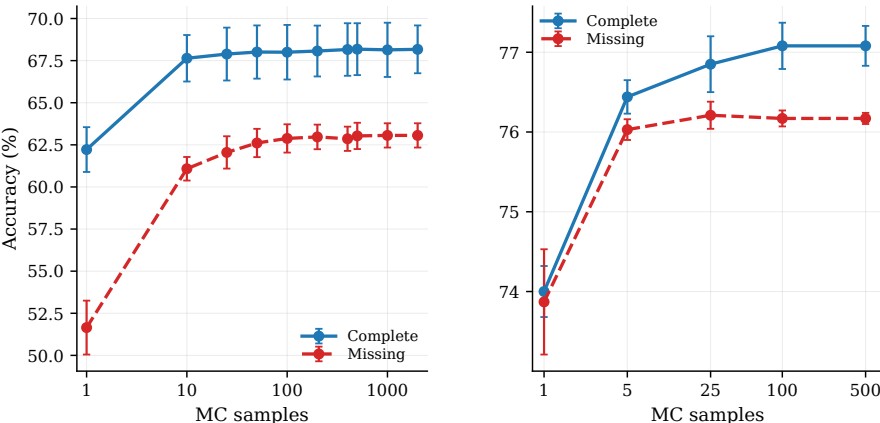

*Figure 18.* **Effect of Monte Carlo (MC) samples on test accuracy** with the audio modality missing (left) and mortality prediction with the time-series modality missing (right). Accuracy does not change much beyond 500 samples, but using more samples reduces variance in the predictive distribution with minimal impact on accuracy.

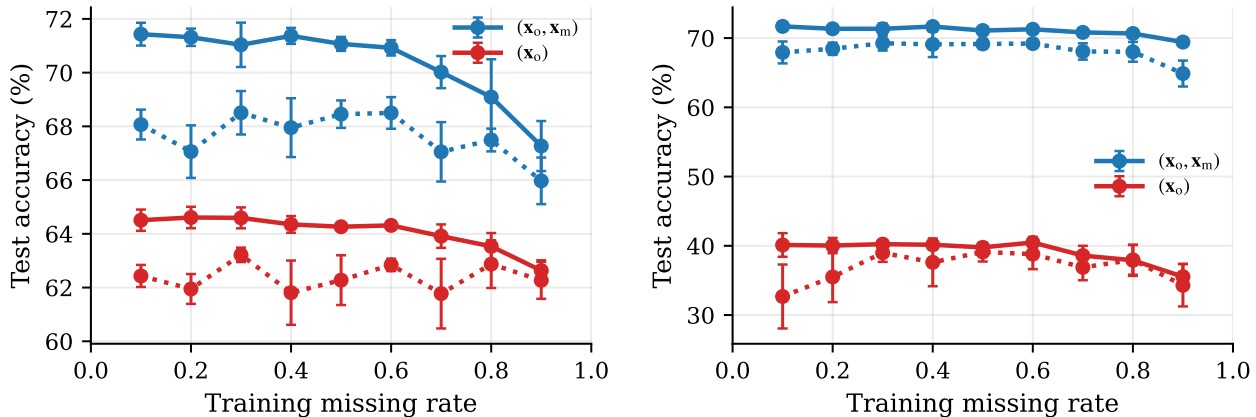

*Figure 19.* **Effect of training missing rate on test accuracy** in complete and missing-modality scenarios, with audio missing on the left and image missing on the right. Dashed lines show PRIMO in the complete and missing modality scenarios, while solid lines show baselines trained with $(\mathbf{x}_o, \mathbf{x}_m)$ or only $(\mathbf{x}_o)$. As the training missing rate increases, the gap between PRIMO and the corresponding baselines decreases. For the image-missing setting, the variance also decreases with increasing missing rate.

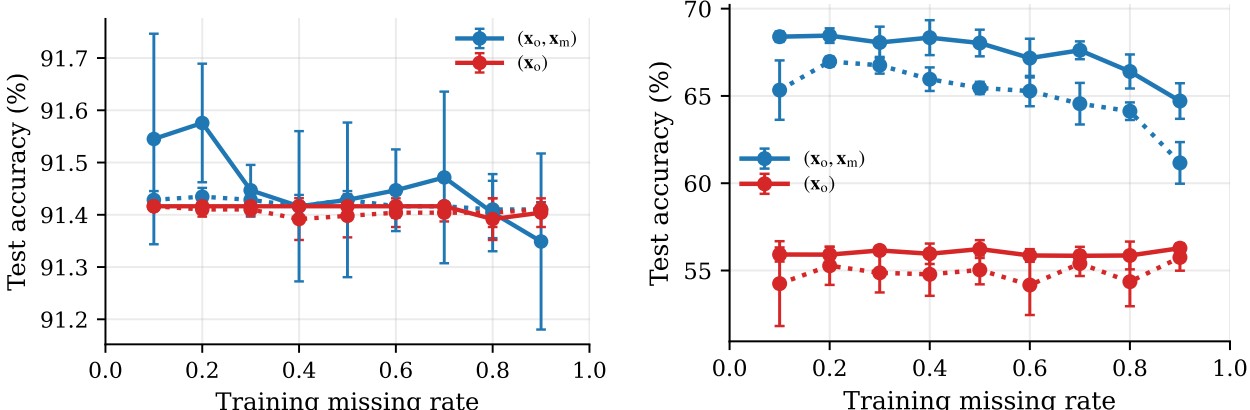

*Figure 20.* **Effect of training missing rate on test accuracy** for ICD-9 group prediction 140-239 (left) and 460-519 (right). Dashed lines show PRIMO in the complete and missing scenarios, solid lines shows baselines trained with $(\mathbf{x}_o, \mathbf{x}_m)$ or only $(\mathbf{x}_o)$. For 140-239, both complete and missing-modality performance remain largely unchanged across training missing rates, suggesting that the task is relatively insensitive to the time-series modality. In contrast, for 460-519, complete-modality performance drops as the training missing rate increases, indicating more reliance on the time-series modality for this task.

