# OpenReview forum: "Characterizing the Predictive Impact of Modalities with Supervised Latent-Variable Modeling"
_ICML.cc/2026/Conference — ICML 2026 regular_

### Official Review · Reviewer_gU7h · 2026-03-07

**Soundness:** 3
**Presentation:** 3
**Significance:** 3
**Originality:** 3
**Overall Recommendation:** 5
**Confidence:** 4

**Summary:**

The paper studies the role of modality variance in multimodal representation learning and introduces PRIMO, a method that regulates the variance of modality-specific latent representations during training. The main idea is that large differences in variance across modalities can lead to unstable multimodal representations and poor robustness when modalities are missing. PRIMO addresses this issue by introducing a regularization mechanism that constrains and balances the variance of modality embeddings in the latent space. The goal is to encourage more stable and comparable representations across modalities, improving robustness in both complete and incomplete modality settings.

**Compliance With Llm Reviewing Policy:**

Affirmed.

**Final Justification:**

All of my concerns have been adequately addressed during the rebuttal phase. In light of the authors’ clarifications, I believe this work meets the standards of the venue and should be accepted.

**Key Questions For Authors:**

The experimental results show that PRIMO consistently achieves worse performance compared to the baseline in both complete and incomplete modality scenarios. For example, on AV-MNIST the accuracy is about 3% lower in the complete setting and between 1% and 3% lower in the incomplete setting. Could the authors clarify what the practical advantage of PRIMO is in these cases?

The experimental evaluation is limited to three datasets, including a synthetic dataset and relatively small benchmarks. Have the authors considered evaluating the method on larger and more complex multimodal datasets to better demonstrate its generalization capabilities?

Finally, the implementation details are only briefly described in the appendix. Could the authors provide more details about the training procedure and hyperparameters to make the method easier to reproduce?

**Limitations:**

yes

**Strengths And Weaknesses:**

* Soundness. The method appears conceptually sound and is based on known ideas related to latent space manipulation and representation regularization. The analysis on modality variance is particularly interesting and helps support the motivation of the work. However, the implementation details are only briefly described and mostly delegated to a small section in the appendix. The models used in the experiments are also very small architectures such as MLPs, GRUs, and LeNet-like networks, which may not be sufficient to fully validate the method. The experimental evaluation is limited. Comparisons with other baselines are only reported for the synthetic XOR dataset, while for the other datasets no comparisons with existing methods are provided. Moreover, PRIMO consistently achieves worse performance than the baseline in both complete and incomplete modality scenarios.

* Presentation. The paper is clearly written and the narrative is easy to follow. The motivation is well presented and the structure of the paper is logical. Nevertheless, there are several repetitions of concepts across sections, especially between the introduction and the method section. In the introduction a first formal definition of the methos is given and then shortly after repeated in the method section.
Some definitions are introduced too quickly. For example, the variance symbol is introduced in the introduction without explicitly stating that it refers to variance (rows 071-075 R).

* Significance. The paper addresses a relevant problem in multimodal representation learning, namely understanding and controlling the statistical properties of modality representations. The analysis of modality variance is interesting and could inspire further work studying the dynamics of multimodal latent spaces. However, the limited experimental evaluation and the use of very small models reduce the strength of the empirical evidence. As a result, the potential impact of the work is currently difficult to fully assess.

* Originality. The idea of controlling the variance of modality representations is interesting, but the proposed method builds on concepts that are already well known in the literature on representation regularization and latent space manipulation. The novelty therefore appears moderate. The most original aspect of the paper is the analysis connecting modality variance and multimodal learning behavior, which helps motivate the proposed method.

---

> ### Author Rebuttal · Authors · 2026-03-26
>
> Dear Reviewer gU7h,
>
> Thank you for your insightful and detailed feedback. We appreciate that you found our proposed framework to be conceptually sound, easy to follow with a well presented motivation. We address your specific concerns below.
>
> > [1] The experimental results show that PRIMO consistently achieves worse performance compared to the baseline in both complete and incomplete modality scenarios. Could the authors clarify what the practical advantage of PRIMO is in these cases?
>
> The goal of our method is not to show an empirical advantage over existing methods but to show the impact of missing modality on the predictions (see Figures 4,5 and 6) , which is not captured in any of the prior data imputation or modality imputation methods. Our objective was to achieve a similar performance to the oracle unimodal and mulitmodal baselines with the introduction of our latent variable framework, while adding this additional capability for multimodal learning.
>
> Further, PRIMO uses hybrid setting which involves a mixture of complete and incomplete examples during training and testing. Contrarily, complete and incomplete baselines represent the oracle performance.
>
> > [2] Comparisons with other baselines are only reported for the synthetic XOR dataset.
>
> First, we want to highlight that the unimodal and multimodal performance we compared with for all the datasets represent the oracle performance. This is the best performance we can achieve when training with a single or both the modalities.
>
> Second, we did not report a comparison on other datasets, because none of these methods can capture how the imputation distribution affects the predictive distribution. Nonetheless, we show a comparison on all the tasks, where we observe that none of these methods consistently match the performance of oracle or PRIMO for both complete and missing modality scenarios.
>
> | Complete     | Audio  | Image  | Mortality         | ICD 140-239           | ICD 460-519           |
> |----------------|-----------------|-----------------|-------------------|-------------------|-------------------|
> | Inter  | $71.14_{0.42}$  | $71.32_{0.30}$  | $77.89_{0.17}$    | $91.37_{0.10}$    | $68.22_{0.52}$    |
> | MVAE           | $66.59_{0.41}$  | $66.52_{0.45}$  | $75.30_{0.78}$    | $90.51_{0.55}$    | $65.54_{0.58}$    |
> | MMVAE          | $66.13_{0.41}$  | $66.31_{0.46}$  | $74.81_{0.96}$    | $90.23_{0.90}$    | $65.30_{0.94}$    |
> | CMMD           | $64.08_{0.23}$  | $40.81_{0.81}$  | $76.32_{0.07}$    | $91.42_{0.00}$    | $56.25_{0.35}$    |
> | LVAE           | $64.08_{0.31}$  | $66.96_{0.78}$  | $75.45_{0.78}$    | $90.39_{0.60}$    | $65.73_{0.76}$    |
> | PRIMO          | $68.17_{1.42}$  | $68.27_{1.35}$  | $77.08_{0.25}$    | $91.41_{0.03}$    | $65.78_{1.08}$    |
>
> | Missing   | Audio  | Image | Mortality       | ICD 140-239         | ICD 460-519         |
> |----------|------------------|-----------------|-----------------|-----------------|-----------------|
> | Unimodal | $64.23_{0.17}$   | $40.36_{0.80}$  | $76.36_{0.01}$  | $91.42_{0.00}$  | $56.22_{0.46}$  |
> | MVAE     | $63.06_{0.37}$   | $38.89_{0.91}$  | $76.25_{0.08}$  | $91.42_{0.00}$  | $55.02_{0.34}$  |
> | MMVAE    | $63.32_{0.30}$   | $38.83_{0.47}$  | $76.27_{0.08}$  | $91.42_{0.0}$   | $55.27_{0.35}$  |
> | CMMD     | $64.08_{0.21}$   | $40.81_{0.81}$  | $76.32_{0.07}$  | $91.42_{0.00}$  | $56.25_{0.35}$  |
> | LVAE     | $62.43_{0.48}$   | $39.01_{0.80}$  | $76.30_{0.05}$  | $91.42_{0.00}$  | $55.16_{0.27}$  |
> | PRIMO    | $63.06_{0.72}$   | $37.58_{1.29}$  | $76.17_{0.07}$  | $91.41_{0.01}$  | $54.95_{1.44}$  |
>
> ---
>
> > [3] Have the authors considered evaluating the method on larger and more complex multimodal datasets to better demonstrate its generalization capabilities?
>
> Evaluation on more complex multimodal datasets with MLLMs is an important direction for future work as the focus of this work was to show the impact of missing modalities on predictions. All prior evaluated baselines (MVAE, MMVAE, CMMD, LVAE) have shown their evaluation on datasets like MNIST, SVHN, FashionMNIST, CelebA. Our work shows that none of these methods work even for a simple synthetic dataset setup like XOR and further scales PRIMO to commonly used multi-modal datasets like AVMNIST and MIMIC-III.
>
> ---
> > [4] Implementation details are only briefly described in the appendix. Could the authors provide more details about the training procedure and hyperparameters to make the method easier to reproduce?
>
> We would appreciate if the reviewer highlight what hyperparameters or training procedures were unclear. We will further release the code in the final version.
>
> ---
>
> Thank you again for your time and efforts in reviewing our paper, and we hope that you will consider raising your score if you find our response satisfactory.
>
> Thank you,
> Authors.

---

> > ### Author Rebuttal · Reviewer_gU7h · 2026-04-03
> >
> > Thank you to the authors for taking the time to address my concerns. Most of the issues have been satisfactorily resolved, so I will increase my score.

---

### Official Review · Reviewer_YExV · 2026-03-09

**Soundness:** 3
**Presentation:** 3
**Significance:** 2
**Originality:** 3
**Overall Recommendation:** 4
**Confidence:** 4

**Summary:**

This paper proposes PRIMO, a supervised latent-variable imputation model for multi-modal prediction with missing modality. Besides prediction, PRIMO also quantifies the predictive impact of any missing modality through a latent variable z, which aims to capture the label-relevant information for the missing modality. The paper introduces a variation score V, which measures the uncertainty of the missing modality to allow analysis of the instance-level impact. Experimental results on both synthetic and real-world multi-modal datasets have shown that PRIMO obtains performance comparable to unimodal baselines when a modality is missing and multimodal baselines when all modalities are available, while still providing insights into the impact of different modalities.

**Compliance With Llm Reviewing Policy:**

Affirmed.

**Final Justification:**

Most of my concerns have been addressed. It would be better to include additional ablation studies in the final version. Hence, I'm keeping my original score.

**Key Questions For Authors:**

- The current paper only focuses on two modalities, with one always observed. How does the method scale to more modalities and more general missingness patterns? What are the specific requirements for the input data in this case?
- How sensitive is the method to the number of Monte Carlo samples used at inference, and why are different numbers of samples used for different datasets?

**Limitations:**

yes

**Strengths And Weaknesses:**

Strengths:
+ The paper studies a meaningful and practical problem in multi-modal learning with missingness, i.e., not only making predictions with missing modality, but also understanding the impact of the missing modality for each instance.
+ The proposed score V provides an intuitive way to quantify and visualize the effect of the missing modality for the output prediction.
+ Beyond accuracy, the paper provides various kinds of visualizations, which help understand how different modalities affect the prediction.

Weaknesses:
- The paper only compared with prior baseline methods on the synthetic dataset. It’s not clear how the proposed method performs relative to existing approaches beyond the complete modality and single modality results.
- The setup in this paper is rather strict. The method is currently limited to only two modalities, with one of them always assumed to be presented. Although the paper mentioned a possible extension to multiple modalities, this assumption is still fairly strong for a real-world dataset. For example, the MIMIC-III experiment uses mean imputation whenever the entire feature set is missing, which makes it is unclear how much this affects the result.
- Minor weakness:
  a) It’s unclear what the small numbers shown in the tables represent.
  b) There are no references or formulas for TVD and DPGMM.
  c) Acronyms such as ECDFs are not introduced.

---

> ### Author Rebuttal · Authors · 2026-03-26
>
> Dear Reviewer YEXv,
>
> Thank you for your insightful feedback. We appreciate that you found our proposed framework to be tackling a meaningful and practical problem. We address your specific concerns below.
>
> > [1] The paper only compared with prior baseline methods on the synthetic dataset. It’s not clear how the proposed method performs relative to existing approaches beyond the complete modality and single modality results.
>
> First, we want to highlight that the unimodal and multimodal performance we compared with for all the datasets represent the oracle performance. This is the best performance we can achieve when training with a single or both the modalities.
>
> Second, we did not report a comparison on other datasets, because none of these methods can capture how the imputation distribution affects the predictive distribution. Nonetheless, we show a comparison on all the tasks, where we observe that none of these methods consistently match the performance of oracle or PRIMO for both complete and missing modality scenarios.
>
> | Complete     | Audio  | Image  | Mortality         | ICD 140-239           | ICD 460-519           |
> |----------------|-----------------|-----------------|-------------------|-------------------|-------------------|
> | Inter  | $71.14_{0.42}$  | $71.32_{0.30}$  | $77.89_{0.17}$    | $91.37_{0.10}$    | $68.22_{0.52}$    |
> | MVAE           | $66.59_{0.41}$  | $66.52_{0.45}$  | $75.30_{0.78}$    | $90.51_{0.55}$    | $65.54_{0.58}$    |
> | MMVAE          | $66.13_{0.41}$  | $66.31_{0.46}$  | $74.81_{0.96}$    | $90.23_{0.90}$    | $65.30_{0.94}$    |
> | CMMD           | $64.08_{0.23}$  | $40.81_{0.81}$  | $76.32_{0.07}$    | $91.42_{0.00}$    | $56.25_{0.35}$    |
> | LVAE           | $64.08_{0.31}$  | $66.96_{0.78}$  | $75.45_{0.78}$    | $90.39_{0.60}$    | $65.73_{0.76}$    |
> | PRIMO          | $68.17_{1.42}$  | $68.27_{1.35}$  | $77.08_{0.25}$    | $91.41_{0.03}$    | $65.78_{1.08}$    |
>
> | Missing   | Audio  | Image | Mortality       | ICD 140-239         | ICD 460-519         |
> |----------|------------------|-----------------|-----------------|-----------------|-----------------|
> | Unimodal | $64.23_{0.17}$   | $40.36_{0.80}$  | $76.36_{0.01}$  | $91.42_{0.00}$  | $56.22_{0.46}$  |
> | MVAE     | $63.06_{0.37}$   | $38.89_{0.91}$  | $76.25_{0.08}$  | $91.42_{0.00}$  | $55.02_{0.34}$  |
> | MMVAE    | $63.32_{0.30}$   | $38.83_{0.47}$  | $76.27_{0.08}$  | $91.42_{0.0}$   | $55.27_{0.35}$  |
> | CMMD     | $64.08_{0.21}$   | $40.81_{0.81}$  | $76.32_{0.07}$  | $91.42_{0.00}$  | $56.25_{0.35}$  |
> | LVAE     | $62.43_{0.48}$   | $39.01_{0.80}$  | $76.30_{0.05}$  | $91.42_{0.00}$  | $55.16_{0.27}$  |
> | PRIMO    | $63.06_{0.72}$   | $37.58_{1.29}$  | $76.17_{0.07}$  | $91.41_{0.01}$  | $54.95_{1.44}$  |
>
>
> ---
> > [2] The method is currently limited to only two modalities.
>
> We want to highlight that no part of our method is constrained by the number of modalities. We simply need a $z$ for each modality and our framework can work as-is for any number of modalities. We used two modalities as that has been a common choice for a lot of the recent multimodal papers [1,2,3,4].
>
> We further emphasize that the objective of our experiments was to not just show that PRIMO works, but to carefully analyze how it works. For instance, for two modalities in MIMIC-III, the importance of time-series modality varies significantly across different tasks (see Figure 8) and different examples (see Figure 9, Figure 13 and Figure 14). We additionally stratify plausible labels by age and chronic conditions in Figure 7, providing insights into the patterns observed in Figure 8.
>
> ---
> [3] a) what the small numbers in the tables. b) no references or formulas for TVD and DPGMM. c) Acronyms such as ECDFs are not introduced.
>
> * Small numbers show standard deviation across 5 runs.
> * TVD refers to total variation distance.
> $\mathrm{TVD}(P,Q)=\sup_A |P(A)-Q(A)|$. For categorical variables,
> $\mathrm{TVD}(P,Q)=\frac{1}{2}\sum_x |P(x)-Q(x)|.$
> * DPGMM refers to Dirichlet Process Gaussian Mixture Model
> * ECDF denotes empirical cumulative distribution function.
>
> ---
>
> [4] How sensitive is the method to the number of MC samples, and why are different numbers of samples used for different datasets?
>
> The accuracy does not change much beyond 500 samples; larger number of samples reduce variance in the predictive distribution with minimal impact on accuracy.
>
> ---
>
> Thank you again for your time and efforts in reviewing our paper, and we hope that you will consider raising your score if you find our response satisfactory.
>
> Thank you,
> Authors.
>
> [1] Tong et al. Cambrian-1: A Fully Open, Vision-Centric Exploration of Multimodal LLMs.
> [2] Gu et al. The Illusion of Readiness: Stress Testing Large Frontier Models on Multimodal Medical Benchmarks.
> [3] Madaan et al. Multi-modal Data Spectrum: Multi-modal Datasets are Multi-dimensional.
> [4] Datalogy AI, DATBENCH: Discriminative, Faithful, and Efficient VLM Evaluations.

---

> > ### Author Rebuttal · Reviewer_YExV · 2026-04-03
> >
> > Thanks for the rebuttal! It addresses most of my concerns. However, it would be better if an ablation study on the MC samples could be added in the final version.

---

> > > ### Author Response · Authors · 2026-04-05
> > >
> > > We want to thank the reviewer for reading our response. We provide an ablation study on MC samples for audio missing and  mortality prediction tasks below.
> > >
> > > | MC samples | acc (complete) | acc (incomplete) |
> > > |-----------|--------------|----------------|
> > > | 1         | $62.22_{1.33}$ | $51.65_{1.60}$ |
> > > | 10        | $67.64_{1.38}$ | $61.08_{0.70}$ |
> > > | 25        | $67.89_{1.57}$ | $62.05_{0.96}$ |
> > > | 50        | $68.01_{1.58}$ | $62.61_{0.84}$ |
> > > | 100       | $68.00_{1.62}$ | $62.88_{0.84}$ |
> > > | 200       | $68.07_{1.51}$ | $62.97_{0.73}$ |
> > > | 400       | $68.16_{1.56}$ | $62.86_{0.72}$ |
> > > | 500       | $68.18_{1.54}$ | $63.03_{0.78}$ |
> > > | 1000      | $68.14_{1.61}$ | $63.06_{0.72}$ |
> > > | 2000      | $68.17_{1.42}$ | $63.06_{0.72}$ |
> > >
> > > | MC samples | acc (complete) | acc (incomplete) |
> > > |-----------|--------------|----------------|
> > > | 1         | $74.00_{0.32}$ | $73.87_{0.66}$ |
> > > | 5         | $76.44_{0.21}$ | $76.03_{0.13}$ |
> > > | 25        | $76.85_{0.35}$ | $76.21_{0.17}$ |
> > > | 100       | $77.08_{0.29}$ | $76.17_{0.10}$ |
> > > | 500       | $77.08_{0.25}$ | $76.17_{0.07}$ |
> > >
> > > We are happy that we fully resolved your concerns and will include this ablation in the final version as well.
> > >
> > > Thank you,
> > > Authors

---

### Official Review · Reviewer_8Pgb · 2026-03-11

**Soundness:** 3
**Presentation:** 4
**Significance:** 4
**Originality:** 3
**Overall Recommendation:** 5
**Confidence:** 3

**Summary:**

This paper addresses a novel and meaningful problem in multimodal learning: making predictions under missing modalities while characterizing the predictive impact of different modalities.
The problem formulation is well motivated and conceptually interesting.
The proposed approach is technically sound, introducing a unified latent-variable framework for both complete and missing modality settings.
Overall, the paper presents a creative and principled attempt to connect missing-modality prediction with modality-impact characterization.

**Compliance With Llm Reviewing Policy:**

Affirmed.

**Key Questions For Authors:**

Please refer to the weaknesses above.

**Limitations:**

The paper emphasizes that PRIMO can achieve performance close to unimodal baselines under missing-modality settings and close to multimodal baselines under complete-modality settings. However, in the XOR experiment, LVAE appears to show a qualitatively similar pattern, with performance also remaining close to the corresponding baselines in both settings. As a result, this experiment does not clearly establish a distinctive empirical advantage of PRIMO over alternative methods that can also operate under both missing- and complete-modality scenarios.

**Strengths And Weaknesses:**

#### Strengths
1. The paper targets a practically relevant yet relatively underexplored setting in multimodal learning, where predictions must remain reliable under missing modalities rather than assuming fully observed inputs.
2. The work goes beyond standard missing-modality prediction by explicitly introducing the goal of characterizing modality predictive impact, which gives the problem additional analytical and practical value.
3. The proposed framework is conceptually elegant: instead of treating missing-modality prediction and multimodal prediction as two separate problems, it unifies both within a single probabilistic formulation, which makes the method technically coherent and easy to interpret.

#### Weaknesses

1. All experiments appear to use a fixed random masking rate of 0.5, but the paper does not examine sensitivity to different missingness ratios. For a method designed to support partial-modality training and inference, robustness across different missingness rates seems both natural and important to evaluate.
2. On AV-MNIST, although the paper reports comparisons against unimodal and multimodal baselines, it does not provide a comparison to multiple relevant methods that can potentially operate across both settings, as was done on XOR. As a result, the empirical comparison on real datasets appears somewhat limited.
3. The XOR experiment provides an intuitive proof-of-concept for the proposed modality-impact analysis. However, since it is based on a very simple two-dimensional synthetic setting, it offers only limited evidence for the general validity of the proposed impact metric. While ground-truth modality impact is difficult to obtain on real datasets, stronger support could still be provided through a broader range of synthetic experiments with more complex and diverse dependency structures.

---

> ### Author Rebuttal · Authors · 2026-03-25
>
> Dear Reviewer 8Pgb,
>
> Thank you for your insightful and detailed feedback. We appreciate that you found our proposed framework to be conceptually elegant and practically relevant. We address your specific concerns below.
>
> ---
> >[1]  The paper does not provide a comparison to alternative methods that can potentially operate across both settings, as was done on XOR.
>
> First, we want to highlight that the unimodal and multimodal performance we compared with for all the datasets represent the oracle performance. This is the best performance we can achieve when training with a single or both the modalities.
>
> Second, we did not report a comparison on other datasets, because none of these methods can capture how the imputation distribution affects the predictive distribution. While LVAE obtained compelling performance on XOR, it uses a single latent variable for both observed and missing covariates. This limitation is crucial because the relationship between the missing modality and predictive distribution is the main focus of our paper.
>
> Nonetheless, we show a comparison on all the tasks, where we observe that none of these methods consistently match the performance of oracle or PRIMO for both complete and missing modality scenarios.
>
> | Complete     | Audio  | Image  | Mortality         | ICD 140-239           | ICD 460-519           |
> |----------------|-----------------|-----------------|-------------------|-------------------|-------------------|
> | Inter (Oracle) | $71.14_{0.42}$  | $71.32_{0.30}$  | $77.89_{0.17}$    | $91.37_{0.10}$    | $68.22_{0.52}$    |
> | MVAE           | $66.59_{0.41}$  | $66.52_{0.45}$  | $75.30_{0.78}$    | $90.51_{0.55}$    | $65.54_{0.58}$    |
> | MMVAE          | $66.13_{0.41}$  | $66.31_{0.46}$  | $74.81_{0.96}$    | $90.23_{0.90}$    | $65.30_{0.94}$    |
> | CMMD           | $64.08_{0.23}$  | $40.81_{0.81}$  | $76.32_{0.07}$    | $91.42_{0.00}$    | $56.25_{0.35}$    |
> | LVAE           | $64.08_{0.31}$  | $66.96_{0.78}$  | $75.45_{0.78}$    | $90.39_{0.60}$    | $65.73_{0.76}$    |
> | PRIMO          | $68.17_{1.42}$  | $68.27_{1.35}$  | $77.08_{0.25}$    | $91.41_{0.03}$    | $65.78_{1.08}$    |
>
> | Missing   | Audio  | Image | Mortality       | ICD 140-239         | ICD 460-519         |
> |----------|------------------|-----------------|-----------------|-----------------|-----------------|
> | Unimodal | $64.23_{0.17}$   | $40.36_{0.80}$  | $76.36_{0.01}$  | $91.42_{0.00}$  | $56.22_{0.46}$  |
> | MVAE     | $63.06_{0.37}$   | $38.89_{0.91}$  | $76.25_{0.08}$  | $91.42_{0.00}$  | $55.02_{0.34}$  |
> | MMVAE    | $63.32_{0.30}$   | $38.83_{0.47}$  | $76.27_{0.08}$  | $91.42_{0.0}$   | $55.27_{0.35}$  |
> | CMMD     | $64.08_{0.21}$   | $40.81_{0.81}$  | $76.32_{0.07}$  | $91.42_{0.00}$  | $56.25_{0.35}$  |
> | LVAE     | $62.43_{0.48}$   | $39.01_{0.80}$  | $76.30_{0.05}$  | $91.42_{0.00}$  | $55.16_{0.27}$  |
> | PRIMO    | $63.06_{0.72}$   | $37.58_{1.29}$  | $76.17_{0.07}$  | $91.41_{0.01}$  | $54.95_{1.44}$  |
>
> ---
>
> > [2] The XOR experiment provides an intuitive proof-of-concept for the proposed modality-impact analysis. However, since it is based on a very simple two-dimensional synthetic setting, it offers only limited evidence for the general validity of the proposed impact metric. While ground-truth modality impact is difficult to obtain on real datasets, stronger support could still be provided through a broader range of synthetic experiments with more complex and diverse dependency structures.
>
> We would be happy to include more synthetic experiments and would appreciate it if the reviewer could suggest specific datasets.
>
> We further want to highlight that our work advances the existing set of dataset imputation and missing modality paper evaluation. All prior evaluated baselines (MVAE, MMVAE, CMMD, LVAE) have shown their evaluation on datasets like MNIST, SVHN, FashionMNIST, CelebA. Our work shows that none of these methods work even for a simple synthetic dataset setup like XOR and further scales PRIMO to real-world multimodal datasets.
>
> ---
>
> > [3] All experiments appear to use a fixed random masking rate of 0.5.
>
> Thank you for the suggestion, we will expand on missing rates in the update.
>
> ---
>
> Thank you again for your time and efforts in reviewing our paper, and we hope that you will consider raising your score if you find our response satisfactory.
>
> Thank you,
> Authors

---

> > ### Author Rebuttal · Reviewer_8Pgb · 2026-04-03
> >
> > Thanks for the detailed response, and I maintain my positive score.

---

### Official Review · Reviewer_tRqU · 2026-03-11

**Soundness:** 2
**Presentation:** 3
**Significance:** 2
**Originality:** 2
**Overall Recommendation:** 4
**Confidence:** 4

**Summary:**

This paper is motivated by a practical challenge caused by the missingness of modalities, and proposed a unified procedure that can handle prediction with either complete or missing modalities. The effectiveness is demonstrated by extensive experiments.

**Compliance With Llm Reviewing Policy:**

Affirmed.

**Final Justification:**

rebuttal resolves many of my questions

**Key Questions For Authors:**

My main concern lies in the fact that cross-modal structures are unexplored in this paper.
It would be very helpful if the following question can be addressed:
1. since the modalities can be highly correlated, I wonder why the "leave-one-out' type of measure would make sense as it can be biased by correlation.
2. if cross-modal prediction model can be trained, I wonder if replacing missing modalities by their synthetic plug-in predictions would further improve the evaluation. Modality impact measure after disentanglement can be discussed.

**Strengths And Weaknesses:**

## Strength

- This paper is well written and well organized. The idea is very neat.
- This paper is well motivated, and the proposed approach has been shown to be effective compared with baselines.

## Weakness

- Despite its being neat, the setting in this paper seems over-simplified in the following sense: since the modalities can be highly correlated, I wonder why the "leave-one-out' type of measure would make sense, as it can be biased by correlation. The modality impact measure after disentanglement can be discussed.

- Related to the modality impact measure:
    - (1) In multi-modal learning, one appealing aspect is that we may retrieve missing information by leveraging cross-modal correlation. However, this paper leaves this unexplored.
    - (2) Also related to the previous point, with the cross-modal structure being unexplored, PRIMO can actually be further improved: if a cross-modal prediction model can be trained, I wonder if replacing missing modalities by their synthetic plug-in predictions would further improve the evaluation.

---

> ### Author Rebuttal · Authors · 2026-03-25
>
> Dear Reviewer tRQu,
>
> Thank you for your insightful and detailed feedback. We appreciate that you found our work to be well-motivated, well-written, well-organized and effective compared to baselines. We address your specific concerns below.
>
> > [1] Despite its being neat, the setting in this paper seems over-simplified in the following sense: since the modalities can be highly correlated, I wonder why the "leave-one-out' type of measure would make sense, as it can be biased by correlation. The modality impact measure after disentanglement can be discussed.
>
> We agree with the reviewer that the leave-one-out measure is not perfect to measure the marginal impact of a modality and can be biased by correlation. Our choice was motivated by recent papers [1,2,3,4]. All these papers have shown that this is a reasonable way to measure the impact of different modalities. This proxy estimates the lower bound to the modality impact and we will clarify this in the revision.
>
> ---
>
> > [2]  Related to modality impact measure: (1) In multi-modal learning, one appealing aspect is that we may retrieve missing information by leveraging cross-modal correlation. However, this paper leaves this unexplored. (2) Also related to the previous point, with the cross-modal structure being unexplored, PRIMO can actually be further improved: if a cross-modal prediction model can be trained, I wonder if replacing missing modalities by their synthetic plug-in predictions would further improve the evaluation.
>
>
> Thank you for the suggestion. PRIMO captures the cross-modal correlation, which can allow us to use the distribution $q(x_m | x_o)$ as a scoring function to retrieve the best ways to fill the missing modality after going through all $x_m$ in the dataset. This however is disadvantageous as we need to store the entire dataset in the memory for retrieval. A mixture of parametric and non-parametric approach would be an important direction for future work.
>
> ---
>
> Thank you again for your time and efforts in reviewing our paper, and we hope that you will consider raising your score if you find our response satisfactory.
>
> Thank you,
> Authors
>
>
> [1]  Tong et al. Cambrian-1: A Fully Open, Vision-Centric Exploration of Multimodal LLMs.
> [2] Gu et al. The Illusion of Readiness: Stress Testing Large Frontier Models on Multimodal Medical Benchmarks.
> [3] Madaan et al. Multi-modal Data Spectrum: Multi-modal Datasets are Multi-dimensional.
> [4] Datalogy AI, DATBENCH: Discriminative, Faithful, and Efficient VLM Evaluations.

---

> > ### Author Rebuttal · Reviewer_tRqU · 2026-04-03
> >
> > Thanks for the reply, especially the related references.
> > Although the correlation remains an issue (e.g., when modalities are simply high-dimensional augmentations from a single modality, the score can exhibit "false positives"), overall, I find the paper insightful and will also raise my score.

---

### Decision · Program_Chairs · 2026-04-30

**Decision:**

Accept (regular)

**Comment:**

The paper proposes a latent variable model for characterising the predictive impact of missing modalities in a supervised setting. The reviewers found that the paper is sound, the presentation is clear, and the method is novel, though it builds on already known concepts in the literature. The reviewers raised several concerns, e.g., about the fixed missing rate, but these were mostly addressed, except for, e.g., evaluating on larger, more complex multimodal datasets, and the potential bias in the conditional variance metric due to what tRqU characterised as the "leave-one-out" approach. Despite the latter being a potential fundamental design issue with the method, the reviewer who raised this concern still finds the paper insightful and weakly recommends acceptance. Given the reviewers' positive assessment and the few outstanding concerns, I recommend accepting the paper.